# Biomedical Visual Instruction Tuning
# with Clinician Preference Alignment

**Hejie Cui**[1]\*, **Lingjun Mao**[3]\*, **Xin Liang**[3], **Jieyu Zhang**[4],
**Hui Ren**[5,6], **Quanzheng Li**[5,6], **Xiang Li**[5,6]†, **Carl Yang**[2]†

[1] Stanford University [2] Emory University [3] University of California, Berkeley
[4] University of Washington [5] Massachusetts General Hospital [6] Harvard Medical School

## Abstract

Recent advancements in multimodal foundation models have showcased impressive capabilities in understanding and reasoning with visual and textual information. Adapting these foundation models trained for general usage to specialized domains like biomedicine requires large-scale domain-specific instruction datasets. While existing works have explored curating such datasets automatically, the resultant datasets are not explicitly aligned with domain expertise. In this work, we propose a data-centric framework, **Biomed**ical **V**isual **I**nstruction **T**uning with Clinician Preference **Al**ignment (BioMed-VITAL), that incorporates clinician preferences into both stages of generating and selecting instruction data for tuning biomedical multimodal foundation models. First, during the generation stage, we prompt the GPT-4V generator with a diverse set of clinician-selected demonstrations for preference-aligned data candidate generation. Then, during the selection phase, we train a separate selection model, which explicitly distills clinician and policy-guided model preferences into a rating function to select high-quality data for medical instruction tuning. Results show that the model tuned with the instruction data from our method demonstrates a significant improvement in open visual chat (18.5% relatively) and medical VQA (win rate up to 81.73%). Our instruction-following data, models, and code are available at `https://BioMed-VITAL.github.io`.

## 1 Introduction

Recent advancements in large pre-trained multimodal models, such as GPT-4V [1], have demonstrated impressive performance on various language and vision tasks. However, when directly applied to specialized domains like biomedicine, these models may fall short due to their primary focus on general usage rather than domain-specific expertise [44, 32]. To bridge this gap and adapt general domain models to specialized domains, researchers have explored various techniques. Instruction tuning has emerged as a promising approach, involving the fine-tuning of large foundation models to follow explicit, natural language instructions [43, 29, 48]. These instructions are composed of task-specific prompts and their corresponding response, enabling the models to learn and generalize to a wide range of tasks within the target domain.

Although instruction tuning has proven to be an effective method for adapting models to target domains and performing various downstream tasks, its success heavily depends on large-scale instruction-following datasets. Curating large-scale instructional datasets in specialized domains, such as biomedicine, can be expensive and time-consuming, often requiring significant domain expertise. Previous work proposes to use strong language models to generate instruction data

---

\*These authors contributed equally to this work.

†co-corresponding: xli60@mgh.harvard.edu, j.carlyang@emory.edu

38th Conference on Neural Information Processing Systems (NeurIPS 2024) Track on Datasets and Benchmarks.

automatically, which effectively reduces the need for extensive manual annotation [36]. Such paradigms have successfully been adopted to adapt general domain models to biomedicine. For example, LLaVA-Med [19] developed a framework to instruction-tune biomedical language-vision models with GPT-4 generated instruction-following data. This approach has achieved impressive performance on open-ended visual chat and visual question answering benchmarks, highlighting the potential of using model-generation data in the biomedical domain.

However, existing methods for automatically curating datasets do not explicitly incorporate clinician preferences, which may result in models producing irrelevant or impractical output, limiting their utility in real-world applications [11]. Yet, aligning domain expertise with the process of instruction-following datasets curation is challenging. First, advanced data generators, such as GPT-4V, are often proprietary and not publicly available for alignment tuning. Second, clinician-annotated preference data in the biomedical domain is limited, further restricting effective preference learning. The combination of model opacity and data scarcity creates a significant bottleneck in developing high-quality, expert-aligned instruction-following data for instruction-tuning. This hinders the development of domain-specific models that can effectively incorporate expert preferences and requirements, ultimately limiting their practical utility and real-world impact.

To tackle this challenge, we propose an effective data-centric approach, BioMed-VITAL, that incorporates clinician preference into the process of automatically curating instruction-following data for biomedical visual instruction tuning. As shown in Figure 1, BioMed-VITAL consists of three stages: (1) data generation with demonstrations, (2) data selection with a preference distilled model, and (3) visual instruction-tuning. In data generation, we strategically sample a diverse set of instructions to collect clinician preferences, which are used as demonstrations for GPT-4V-based instructional data generation, guiding the data generation toward producing more clinically relevant and useful instruction-following examples. In the data selection stage, we train a data selection model that distills a mixture of preferences from clinician-annotated and model-annotated data guided by clinician-curated criteria. This model is then used to rank the generated data samples, and the top-ranked samples are selected for visual instruction-tuning.

The contributions of this work are summarized as follows:

- We introduce a data-centric framework BioMed-VITAL, which generates and selects instruction-following data aligned with clinician preference for visual instruction tuning. Evaluation indicates an improved data quality and our instruction-tuned models remarkably improve in both open visual chat (18.5% relatively) and three biomedical VQA benchmarks (win rate up to 81.73%).
- We propose a paradigm involving clinician preference during generation and an effective data selection model based on a mixture of preferences. It is shown that our distilled data selection model excels in matching human preferences compared with judgments of GPT-4.
- To facilitate further study, we release $80k$ clinician preference-aligned instruction-following datasets generated and selected from ours, along with the models instruction-tuned based on them. All resources are publicly available on the website `https://BioMed-VITAL.github.io`.

## 2 Background

**Instruction-Tuning.** Instruction tuning has become an effective method for adapting pre-trained language models to a wide range of natural language tasks [50, 42, 41, 45, 12, 31, 35, 6] by providing task-specific instructions and examples. This approach has been further explored in studies like FLAN-T5 [8], LLaMA [37], and LLaMA2 [38], which enables models to understand and follow task-specific instructions without extensive task-specific fine-tuning. Recently, using strong language models to generate instruction data automatically has been proposed to train a high-quality instruction-following model under an academic budget [33, 36, 25]. For example, Stanford Alpaca [36] instruction-tuned LLaMA using `text-davinci-003`-generated instruction-following datasets and achieved competitive performance on various NLP tasks.

**Vision-Language Foundation Models in Biomedical Domain.** General vision-language foundation models have achieved remarkable success across various domains. Researchers in biomedicine have been actively exploring the adaptation of vision-language foundation models to tackle domain-specific tasks [30, 2, 15, 34]. However, effectively adapting vision-language foundation models to specialized domains such as the biomedical presents challenges, particularly due to limited training data. To overcome this challenge, our work aims to establish a data-centric method that aligns domain

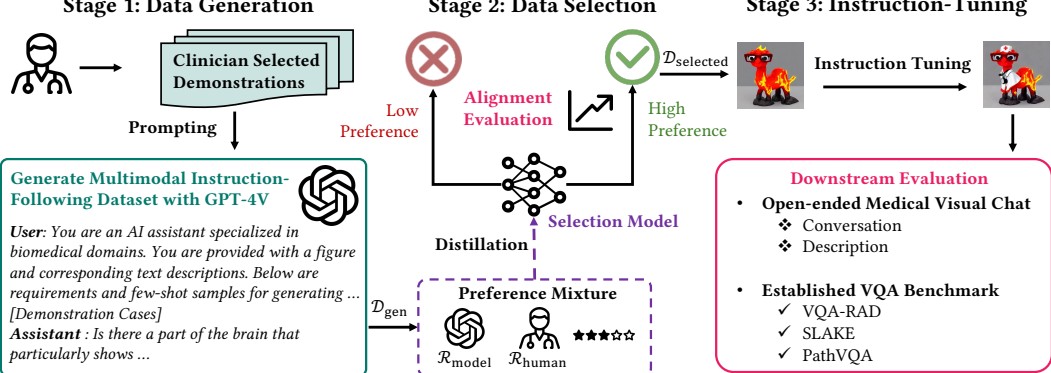

Figure 1: Overview of **Biomed**ical **V**isual **I**nstruction **T**uning with Clinician Preference **Al**ignment (BioMed-VITAL). Clinician preferences are infused in the 1. data generation and 2. selection stages.

expertise from clinicians with the instructional data for instruction-tuning, which generates and selects instruction-following datasets that are aligned with clinician preference.

# 3 Clinician-Aligned Biomedical Visual Instruction Tuning

Figure 1 presents an overview of the proposed framework BioMed-VITAL, consisting of three stages: (1) data generation with diverse expert-selected demonstration, (2) data selection with a distilled selection model trained with mixed preferences, and (3) instruction tuning to adapt a general multimodal model for biomedical tasks. The output from the framework includes a clinician preference-aligned instruction-following dataset $\mathcal{D} = \{(I_i, C_i, \mathbf{Q}_i, \mathbf{A}_i)\}_{i=1}^{N}$ and instruction-tuned models based on it. $I_i$ represents the $i$-th biomedical image; $C_i$ is the caption and inline-mentions associated with the $i$-th image; $\mathbf{Q}_i = \{\mathcal{Q}_{ij}\}_{j=1}^{n_i}$ contains $n_i$ instructions, where $j$ represents the $j$-th instruction for the $i$-th image-text sample; $\mathbf{A}_i = \{\mathcal{A}_{ij}\}_{j=1}^{n_i}$ contains $n_i$ responses, each corresponding to $\mathcal{Q}_{ij}$; and $N$ is the total number of samples in the dataset.

## 3.1 Stage 1: Data Generation with Diverse Expert-Selected Demonstration

Large pre-trained models have shown strong in-context learning capabilities by learning from a few presented examples and mimicking when generating responses. In BioMed-VITAL, we use the GPT-4V model as the generator. To incorporate clinician preference into the data generation process, we first select a diverse set of samples for clinicians to annotate. Clinician-selected QA pairs are used as few-shot demonstrations for GPT-4V to generate instruction-following data at scale.

**Diverse few-shot demonstration selection.** We employ a strategic sampling approach to ensure the diversity and representatives of the demonstrations for the generator. For each sample $(I_i, C_i)$ in the dataset, the image and text representations are extracted using BiomedCLIP [47], then we perform K-means clustering on these representations to cluster the samples into $K$ distinct categories, denoted as $\mathcal{D}_1, \mathcal{D}_2, ..., \mathcal{D}_K$. From these clusters, we uniformly select a subset $S = (I_i, C_i)_{i=1}^{M}$ with total $M$ samples that have relatively complex captions and inline mentions. For each selected sample $(I_i, C_i) \in S$, we use GPT-4V to generate conversations that present instructions $\mathcal{Q}_i$ and two candidate responses $A_i^1, A_i^2$ for each instruction [3]. During the human annotation, each clinician is presented with a carefully selected set of $M$ questions with two response candidates and asked to choose the preferred one $A_{ij}^{\text{pref}}$ between the two, select both if two responses are equally good, or deselect both to drop this instruction [4]. Three clinicians participated in our annotation. The Fleiss' kappa on the three clinician annotations is 0.736, indicating a good agreement among the three annotators. For these cases where disagreement happened, we applied majority voting to make the final decision. The resulting annotation $\mathcal{R}_{\text{human}}$ contains the selected preferences from clinicians.

---

[3]The prompt can be found in Appendix B Figure 6.

[4]See Appendix B Figure 7 for the annotation protocol.

**Instruction-following data generation with GPT-4V.** Using the clinician-selected data, we employ GPT-4V as the generator to simulate the instructional dataset. During each API call, we randomly select 2 samples for each of the 5 modalities from $\mathcal{D}_{\text{pref}}$ as few-shot demonstrations and append them to the language prompts. The full prompt can be referred to in Appendix A Figure 5. Compared with previous methods, our generated dataset $\mathcal{D}_{\text{gen}} = \{(I_i, C_i, \mathbf{Q}_i, \mathbf{A}_i)\}_{i=1}^{N}$ incorporate visual input and is further guided with selected clinician demonstrations.

## 3.2 Stage 2: Distilling Mixed Clinician Preference for Data Selection

While $\mathcal{D}_{\text{gen}}$ is directly usable to instruction-tune, it may still include samples that can introduce noise or bias or are irrelevant to the real needs of clinicians. In the second stage of BioMed-VITAL, we train a data selection model that learns to select instruction data aligned with expert preference.

**Preference data from two resources.** Collecting human preference data from domain experts such as clinicians is expensive and time-consuming. Thus, the available annotation data is usually on a small scale. A recent paradigm involves using LLMs as judges, which have been shown to match human preferences effectively [51]. We consider a data mixing schema to distill preference into a local model for data selection. Our preference data comes from two resources, from humans and from models: (1) human preference from the preference annotation $\mathcal{R}_{\text{human}}$ in stage 1, where each question $\mathcal{Q}_{ij}$ is paired with two candidate answers $A_{ij}^1, A_{ij}^2$, with $A_{ij}^{\text{pref}}$ annotated as the preferred one. (2) model-based preference: to generate reliable model-based ratings, we first collect a set of clinician-curated factors for data quality evaluation, such as missing information, recognition errors, lack of medical precision, insufficient depth, valueless questions, etc. With these clinician-curated criteria, we use GPT-4V as a judge to score a randomly sampled set of data from 0 to 10. The detailed prompt can be referred to in Appendix B Figure 8. The resulting self-evaluated ratings, $\mathcal{R}_{\text{model}}$, provide additional preference data and address the scalability issue related to human annotation.

**Distill clinician preference to a selection model.** Next, we train a data selection model with the preference data, which is designed to identify and remove low-quality samples from the generated dataset and preserve only the most accurate and clinically relevant examples for instruction tuning. We use BiomedCLIP [47] as the backbone, followed by an MLP head to perform binary prediction tasks on good/bad ratings of data samples. Pairwise ranking loss is used as the training objective: given a pair of candidate samples $x_i$ and $x_j$, along with their corresponding annotated preferences $\mathcal{R}_i$ and $\mathcal{R}_j$, the objective is formulated as a pairwise classification:

$$\mathcal{L}_Q = -z_i \log \sigma \left( f(x_i) \right) - z_j \log \sigma \left( f(x_j) \right), \tag{1}$$

where $\sigma$ represents the sigmoid function, and $f(\cdot)$ denotes the rating function learned by the model. The values of $z_i$ and $z_j$ are determined by comparing the preference annotation:

$$(z_i, z_j) = \left\{ \begin{array}{ll} (1,0), & \mathcal{R}_i \geq \mathcal{R}_j \\ (0,1), & \mathcal{R}_i < \mathcal{R}_j \end{array} \right. . \tag{2}$$

By minimizing the pairwise classification loss, the data selection model learns to predict the likelihood of a sample being labeled as 1 within a sampled pair, by assigning higher scores to samples with higher preference and lower scores to samples with lower preference.

**Preference mixing strategy during training.** We mix two sources of preference data in each batch during training. In Eq (1), each $x_i$ and $x_j$ can be either human-annotated preferences from $\mathcal{R}_{\text{human}}$, or two samples with model-based ratings $\mathcal{R}_i$ and $\mathcal{R}_j$ from $\mathcal{R}_{\text{model}}$. To address the scalability difference between the two resources, we introduce an adaptive contribution mechanism by incorporating an adjustable sample weight $w_{i,j}$ into Eq (1):

$$\mathcal{L}_Q = -w_{i,j} \left( z_i \log \sigma \left( f(x_i) \right) + z_j \log \sigma \left( f(x_j) \right) \right), \tag{3}$$

where $w_{i,j}$ allows for an adjustable preference contribution from human or model during training.

**Data selection with distilled selection model.** We apply the trained data selection model to the generated dataset $\mathcal{D}_{\text{gen}}$ and observe F1@K and Precision@K curves to determine the threshold for data selection. To balance data quality and diversity, we first cluster all the data samples into K groups and uniformly select top-ranked data in each group to compose the final instruction-following dataset, denoted as $\mathcal{D}_{\text{selected}}$, which contains the most informative, accurate, and clinically relevant examples. More empirical decisions during selection are discussed in Section 4.2.

### 3.3 Stage 3: Instruction-Tuning

Following LLaVA-Med [19], we continue training the LLaVA [26, 24] model on our curated instruction-following dataset $\mathcal{D}_{\text{selected}}$. The instruction tuning objective for model $\theta$ is to minimize the negative log-likelihood of the target $\mathbf{A}_i$ given input image $I_i$, caption $C_i$, question $\mathbf{Q}_i$,

$$\mathcal{L}_{IT} = - \sum_{i=1}^{|\mathcal{D}_{\text{selected}}|} \log p(\mathbf{A}_i | I_i, C_i, \mathbf{Q}_i, \theta). \tag{4}$$

## 4 Experiments

### 4.1 Dataset and Experiment Details of BioMed-VITAL

We follow the setup of Li et al. [47] and utilize image-text pairs from the PMC-15M dataset [47] to generate multi-round QA instructional data. For the data generator, we utilize `gpt-4-vision-preview` API on Azure OpenAI. For the diverse few-shot demonstration selection, we set $K$ to 60 and $M$ to 300 for simplicity. For the data selector, we use BiomedCLIP [47], which is trained for 6 epochs with a learning rate of 1e-4. For instruction-tuning, we use `llava-v1.5-13b` as the backbone. Following the LLaVA-Med approach [19], the model is first trained with biomedical concept alignment; subsequently, it is instruction-tuned using the selected dataset from the second stage, utilizing a multi-turn dialogue setup [25]. The instruction-tuning process is carried out for 3 epochs with a learning rate of 2e-5, trained and tested with 2 NVIDIA A100 GPUs.

### 4.2 Alignment Evaluation of the Data Selection Model

Table 1: Varying preference mixture strategy.

| Mixture Strategy | Rank-based Metrics (%) | | | |
|---|---|---|---|---|
| | ACC ↑ | AUC ↑ | MR ↓ | MAP ↑ |
| only $\mathcal{R}_{\text{human}}$ | 55.89 | 55.99 | 46.91 | 56.21 |
| only $\mathcal{R}_{\text{model}}$ | 54.76 | 54.64 | 47.67 | 55.25 |
| mix, $w_{\mathcal{R}_{\text{human}}}/w_{\mathcal{R}_{\text{model}}} = 1$ | 61.61 | 61.90 | 44.04 | 62.22 |
| mix, $w_{\mathcal{R}_{\text{human}}}/w_{\mathcal{R}_{\text{model}}} = 5$ | 58.63 | 58.22 | 45.87 | 62.29 |
| mix, $w_{\mathcal{R}_{\text{human}}}/w_{\mathcal{R}_{\text{model}}} = 10$ | 59.38 | 59.14 | 45.39 | 59.20 |
| mix, $w_{\mathcal{R}_{\text{human}}}/w_{\mathcal{R}_{\text{model}}} = 50$ | 59.67 | 59.80 | 45.09 | 63.27 |
| mix, $w_{\mathcal{R}_{\text{human}}}/w_{\mathcal{R}_{\text{model}}} = 100$ | 62.05 | 62.30 | 43.84 | 61.63 |
| mix, $w_{\mathcal{R}_{\text{human}}}/w_{\mathcal{R}_{\text{model}}} = 200$ | 60.91 | 61.23 | 44.37 | 59.55 |
| mix, $w_{\mathcal{R}_{\text{human}}}/w_{\mathcal{R}_{\text{model}}} = 300$ | 63.64 | 63.12 | 43.43 | 63.00 |
| mix, $w_{\mathcal{R}_{\text{human}}}/w_{\mathcal{R}_{\text{model}}} = \mathbf{400}$ | **66.72** | **66.32** | **41.83** | **64.47** |
| mix, $w_{\mathcal{R}_{\text{human}}}/w_{\mathcal{R}_{\text{model}}} = 500$ | 62.85 | 63.06 | 43.46 | 65.00 |
| mix, $w_{\mathcal{R}_{\text{human}}}/w_{\mathcal{R}_{\text{model}}} = 600$ | 56.30 | 56.07 | 46.95 | 60.25 |

**Preference mixture.** In our experiment, we train the selection model with a proportion of 1:400 of human and model preferences to reflect the scalability gap. To find the optimal balance between the two resources, we adjust the ratio of sample weight for human-annotated and model-generated preference $w_{\mathcal{R}_{\text{human}}}/w_{\mathcal{R}_{\text{model}}}$ when applying Eq (3) to train the selection model and observe the performance with varying mixture strategies on a clinician-annotated test set. We also conducted experiments to train the data selection model using only human-generated preference $\mathcal{R}_{\text{human}}$ or only model-generated ratings $\mathcal{R}_{\text{model}}$. The results are summarized in Table 1.

It shows that when trained with only $\mathcal{R}_{\text{human}}$ or $\mathcal{R}_{\text{model}}$, the selection model's performance is inferior to the stratified mixture of both preference data. This indicates that while high-quality, the limited amount of annotations is insufficient for robust data selection model training. In comparison, stratified mixing of $\mathcal{R}_{\text{human}}$ and $\mathcal{R}_{\text{model}}$ significantly improves performance compared to using only $\mathcal{R}_{\text{model}}$. The best performance is achieved with a ratio of 400, which happens to balance the contribution of human and model-annotated preference in the total training loss. These findings strongly support our approach of preference mixing strategy, which effectively balances the high-quality but limited clinician annotations with scalable model-based annotations, resulting in a more accurate selection model that successfully distillates clinician preferences while minimizing human effort in annotation.

**Alignment with human preference versus GPT-4.** To compare the preference evaluation ability of our trained data selection model versus the GPT-4 model, we calculate the correlation between the ratings generated from both models with gold clinician-annotated preference. The results in Figure 2 (left panel) indicate a better alignment of our trained selection model over the GPT-4 model.

**Selecting top K ranked samples.** We observe the F1 and Precision performance curves on the ranking list from the score model by varying the top K percentiles to determine the optimal proportion of top-ranked data to select. As illustrated by Figure 2 (right panel), we identify three critical percentiles: 10%, 50%, and 80%, where the performance either reaches a local peak or plateaus afterward, indicating that further incorporating data on the ranking list would not yield significant improvements. The top 50% and the top 80% are selected because they demonstrate similar F1 scores and precision

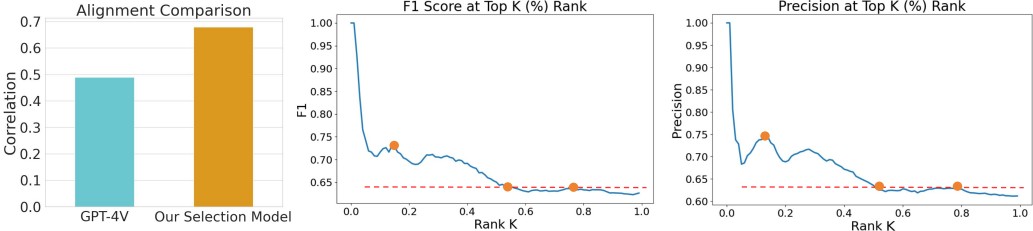

Figure 2: Left: Comparison of human preference alignment between GPT-4V and our selection model. Right: F1 and precision for varying top K percentile samples ranked by the selection model.

when compared to human annotations. This allows us to understand the influence of the scaling law in model training. Additionally, we include the top 10% threshold as it represents a subset of data that strikes a balance between quality and quantity to help better understand the effectiveness of data selection. Consequently, we select datasets corresponding to these critical percentiles for visual instruction tuning, ensuring that the models learn from high-quality, clinician-preferred samples.

## 4.3 Downstream Evaluation 1: Open-Ended Medical Visual Chat

To evaluate the model's ability to engage in dialogue-like interactions and provide coherent responses, we evaluate the model with open-ended visual chat, where the trained language models are prompted to respond to given questions based on the provided images and texts in a multi-round manner.

**Dataset and evaluation paradigm.** For the evaluation dataset, we use 50 unseen image and caption pairs with 193 question-answer pairs collected by the LLaVA-Med [19] authors. The questions are divided into two types: (1) Conversation questions, which require the model to engage in dialogue-like interaction, understand the context and provide relevant responses. For example, given an image of a chest X-ray, a conversation question might ask, "What abnormalities do you see in this X-ray image?" (2) Description questions, which focus on detailed descriptions or explanations based on visual and textual input. For instance, a description question for a histology image could be, "Describe the morphological features of the cells in this histology slide."

Open-ended visual chat can be challenging to evaluate, with traditional NLP metrics insufficient to capture the semantic and higher-order abstract text aspects. Recent studies have explored using LLMs as evaluators, demonstrating their greater resilience compared to metric-based approaches [3, 7, 27]. Following these, we use GPT-4V as the evaluator. A reference prediction is first generated based on the input context and the given question, which is then provided to assess the responses from various trained models by assigning a relative score on a scale from 1 to 10. A higher score indicates that the model's response is more accurate, relevant, and coherent with respect to the reference prediction.

Table 2: Performance comparison of the instruction-tuned models on open-ended biomedical visual chat. The number followed by "#: " represents the number of testing samples in this category. In the following experiments, $N$ is the number of QA pairs of 60k images.

| Model | Data Size | Question Types | | Domains | | | | | Overall |
| | | Conversation (#:143) | Description (#: 50) | CXR (#: 37) | MRI (#: 38) | Histology (#: 44) | Gross (#: 34) | CT (#: 40) | (#: 193) |
|---|---|---|---|---|---|---|---|---|---|
| LLaVA-Med | $N$ | 58.53 | 56.16 | 43.97 | 51.19 | 60.01 | 86.49 | 50.63 | 57.92 |
| BioMed-VITAL | Top 10% $*N$ | 64.11 | 60.05 | 56.35 | 52.57 | 59.02 | 87.60 | 62.82 | 63.06 |
| BioMed-VITAL | Top 50% $*N$ | 65.95 | 64.26 | 55.75 | 55.57 | 60.96 | 94.06 | 64.70 | 65.51 |
| BioMed-VITAL | Top 80% $*N$ | 68.50 | **67.65** | 55.24 | **58.73** | 62.65 | **101.88** | **67.05** | 68.28 |
| BioMed-VITAL | $N$ | **69.73** | 65.51 | **59.22** | 57.39 | **67.15** | 99.26 | 63.63 | **68.63** |
| *Model Ablation* | | | | | | | | | |
| BioMed-VITAL[A0] | $N$ | 65.38 | 60.63 | 63.48 | 53.82 | 57.32 | 92.30 | 58.16 | 64.15 |
| BioMed-VITAL[A1] | $N$ | 67.82 | 59.48 | 59.68 | 53.98 | 60.34 | 97.89 | 60.74 | 65.66 |
| BioMed-VITAL[A2] | $N$ | 67.53 | 62.78 | 60.64 | 54.62 | 61.07 | 98.27 | 61.21 | 66.30 |

**Model variants.** In addition to comparing our model with the LLaVA-Med baseline, we further investigate the influence of the selected data size on instruction tuning performance and conduct a model ablation study. ♦ To study the impact of data size, we instruction-tune three additional models using datasets selected from the ranking list at three critical percentiles: 10%, 50%, and 80%, as described in Section 4.2 and illustrated in Figure 2. ♦ For the model ablation study, we include three

Table 3: Statistics of the benchmark datasets for downstream evaluation on biomedical VQA.

| Dataset | VQA-RAD | | SLAKE | | | PathVQA | | |
|---|---|---|---|---|---|---|---|---|
| | Train | Test | Train | Val | Test | Train | Val | Test |
| # Images | 313 | 203 | 450 | 96 | 96 | 2,599 | 858 | 858 |
| # QA Pairs | 1,797 | 451 | 4,919 | 1,053 | 1,061 | 19,755 | 6,279 | 6,761 |
| # Open | 770 | 179 | 2,976 | 631 | 645 | 9,949 | 3,144 | 3,370 |
| # Closed | 1,027 | 272 | 1,943 | 422 | 416 | 9,806 | 3,135 | 3,391 |

variants based on the full BioMed-VITAL model: BioMed-VITAL$^{A0}$, which does not incorporate clinician preference alignment in either stage; BioMed-VITAL$^{A1}$, which only includes the first stage of clinician-selected demonstrations; and BioMed-VITAL$^{A1}$, which only incorporates the second stage of preference distillation. The results of these investigations are summarized in Table 2. Note the numbers are relative scores calculated following LLaVA-Med [19]. When comparing a response generated by a model to a reference answer from GPT-4V, the evaluator is asked to provide scores for both, and a relative score is calculated by dividing the model's score by the reference's score.

**Result discussion.** For the three-dimensional comparison:

- Baseline comparison: BioMed-VITAL and all its variants consistently outperform the compared method. Even with only the top 10% of selected data, the BioMed-VITAL model surpasses the baseline model trained on the full dataset of size $N$ in both question types, highlighting the effectiveness of our data-centric framework.
- Data size study: When varying the top-ranked percentiles in the data selection process, increasing the dataset size generally improves model performance. Notably, our models trained with fewer data (i.e., 50% and 80% of the dataset) outperform the BioMed-VITAL$^{A0}$ and BioMed-VITAL$^{A1}$ models, which are trained on the full data size $N$ without data selection. This finding suggests that the second-stage data selection leads to more efficient and effective model tuning, as it focuses on the most informative and relevant examples.
- Model ablation study: Comparing the three model ablations with the full BioMed-VITAL model, we observe that incorporating clinician preference infusion in both the data generation and selection stages leads to improved performance compared to the base model. The full BioMed-VITAL model achieves the best performance, revealing the effectiveness of combining both alignment stages to achieve optimal results. This finding underscores the importance of considering clinician preferences throughout the entire data-centric framework for biomedical visual instruction tuning.

### 4.4 Downstream Evaluation 2: Performance on Established VQA Benchmarks

**Dataset details.** We train and evaluate BioMed-VITAL on three widely used biomedical visual question answering benchmarks [19, 39, 49]. The statistics of the datasets are shown in Table 3.

- VQA-RAD [17] is a dataset containing 3,515 question-answer pairs created by medical professionals, along with 315 radiology images. Each image is linked to several questions, which are categorized into 11 types, including abnormality, attribute, modality, organ system, color, counting, object/condition presence, size, plane, positional reasoning, and others. The dataset features a balanced mix of closed-ended (yes/no) and open-ended (one-word or short phrase) answers.
- SLAKE [23] is a comprehensive medical visual question-answering dataset with knowledge-enhancement features. It contains radiology images and diverse question-answer pairs annotated by experienced physicians. SLAKE covers a wide range of modalities and human body parts, such as the brain, neck, chest, abdomen, and pelvic cavity.
- PathVQA [13] focuses on pathology images. Each image is associated with multiple questions that cover various aspects, such as location, shape, color, and appearance. The questions in PathVQA include open-ended questions (e.g., why, what, how, where) and closed-ended questions.

**Experimental details.** For each benchmark, the model is fine-tuned for 15 epochs with a learning rate of 2e-5. To account for the open-ended nature and expressive diversity of language generation, we report both metrics-based performance and an additional model-based win rate performance. The win rate performance provides a complementary perspective on the model's ability to generate accurate and relevant responses compared to the baseline.

**Metric performance.** To evaluate the performance metrics, we follow the practice of Li et al. [19] and use accuracy for closed-set questions and recall (the ratio of ground-truth tokens appearing in the generated response) for open-set questions. Table 4 summarizes the metric performance of

Table 4: Metric performance of BioMed-VITAL and compared methods on three VQA benchmarks. Models based on LLaVA are trained with 7b/13b backbone and training sample size of 60K/150K. The largest set 150K combines 10K and 60K provided by LLaVA-Med, plus our curated 80K samples.

| Model | VQA-RAD | | | SLAKE | | | PathVQA | | |
|---|---|---|---|---|---|---|---|---|---|
| | Ref | Open | Closed | Ref | Open | Closed | Ref | Open | Closed |
| *Supervised fine-tuning results from models based on LLaVA (model size, training sample size)* | | | | | | | | | |
| LLaVA (7B) | | 50.00 | 65.07 | | 78.18 | 63.22 | | 7.74 | 63.20 |
| LLaVA-Med (7B, 60k) | | 61.52 | 84.19 | | 83.08 | 85.34 | | 37.95 | 91.21 |
| LLaVA-Med (13B, 60k) | | 64.58 | 77.94 | | 84.97 | 85.58 | | 38.82 | 92.39 |
| BioMed-VITAL (7B, 60k) | | 63.46 | 84.71 | | 85.41 | 87.26 | | 38.96 | 92.39 |
| BioMed-VITAL (13B, 60k) | | 64.88 | 84.55 | | 87.82 | 86.54 | | 39.71 | 91.41 |
| BioMed-VITAL (13B, 150k) | | **69.72** | **84.86** | | **91.69** | **90.70** | | **39.89** | **92.42** |
| *Literature-reported results from representative SoTA methods* | | | | | | | | | |
| MMQ [9] | 53.70 | | 75.80 | | | | 13.40 | | 84.00 |
| Prefix T. Medical LM [40] | | | | 84.30 | | 82.01 | 40.00 | | 87.00 |
| PubMedCLIP [10] | 60.10 | | 80.00 | 78.40 | | 82.50 | | | |
| BiomedCLIP [46] | 67.60 | | 79.80 | 82.05 | | 89.70 | | | |
| M2I2 [21] | 66.50 | | 83.50 | 74.70 | | 91.10 | 36.30 | | 88.00 |
| MUMC [20] | 71.50 | | 84.20 | 81.50 | | 81.50 | 39.00 | | 65.10 |
| M3AE [5] | 67.23 | | 83.46 | 80.31 | | 87.82 | | | |
| CoQAH [16] | 30.20 | | 67.50 | 42.50 | | 73.90 | | | |
| PMC-CLIP [22] | 67.00 | | 84.00 | 81.90 | | 88.00 | | | |

BioMed-VITAL compared to models based on LLaVA, as well as literature-reported results from representative state-of-the-art (SoTA) methods for reference[5]. Among the supervised fine-tuning models based on LLaVA, BioMed-VITAL consistently outperforms the other two, particularly on open-type questions, with the 150k trained model achieving the best. When comparing ours to those reported in the literature from previous methods, it is important to note that some prior methods formulate the problems as classification tasks among answer candidates in the training set, which does not meet the real-world need for open-ended QA. Additionally, some studies report metrics on the open set using different calculations, leading to inconsistencies in comparison. We follow the practice of Li et al. [19] and present the numbers from prior work only as a reference for the open set while including metrics on the closed set for comparison. The results demonstrate that BioMed-VITAL achieves leading performance in most cases, even when compared to methods that employ classification set up for QA despite BioMed-VITAL being in an open, generative manner.

**Varying vision-language model backbone and model sizes.** We conducted additional experiments with recent SoTA language-vision models, including LLaVA-OneVision [18] and InternVL-1.5 [4], trained on our generated instruction data, and evaluated on three VQA benchmarks. The results are summarized below in Appendix C Table 5. The key findings include: (1) Fine-tuning with our instruct data significantly improves model performance across all three benchmarks, demonstrating the effectiveness of our framework in generating helpful training data. (2) Our approach shows consistent improvement in both open and closed categories compared to fine-tuning with LLaVA-Med datasets, highlighting the benefits of clinician alignment. (3) Performance gains are observed across different model architectures, indicating the generalizability and robustness of our approach.

We also conducted experiments using models of various sizes (7B and 13B parameters). As shown in Appendix C Table 6, our framework consistently outperforms both LLaVA and LLaVA-Med across tasks and model sizes, particularly notable in the open-ended questions for all datasets. This underscores our framework's generalizability across model sizes.

**Win rate performance.** Recent studies in visual question-answering have highlighted the limitations of token-matching metric evaluation for open-ended language generation tasks and have proposed leveraging model-based win rate evaluation instead [28, 14]. In line with these insights, we adopted a reference-guided win rate evaluation, where GPT-4V is employed as an impartial judge to assess the quality of the responses provided by two compared models. The detailed prompt for win rate evaluation on VQA benchmarks is shown in Appendix E Figure 9. By considering the ground-truth reference, GPT-4V determines which model provides the more accurate and relevant answer, offering a comprehensive evaluation of the models' performance in reponse generation.

---

[5]Details of the compared SoTA methods can be referred to in Appendix D.

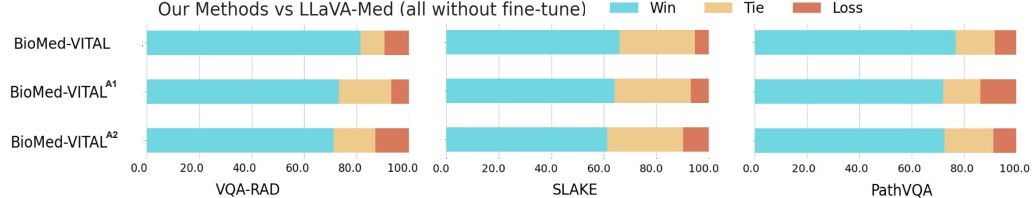

Figure 3: Win rate performance of BioMed-VITAL and its variants compared with LLaVA-Med.

As shown in Figure 3, BioMed-VITAL and its variants BioMed-VITAL[A1] and BioMed-VITAL[A2] outperform the LLaVA-Med baseline and achieve significantly higher win rates up to 81.73%. It is worth noting that the full model consistently performs the best compared to the two ablations, indicating the effectiveness of the clinician preference alignment during both the data generation and selection phases. Between the two model variants, BioMed-VITAL[A1], which only incorporates clinician alignment in the data generation phase, performs slightly better than BioMed-VITAL[A2], which only incorporates clinician alignment in the data selection phase. This finding indicates the greater impact of the generation phase on clinician preference alignment than the selection phase.

## 4.5 Case Study

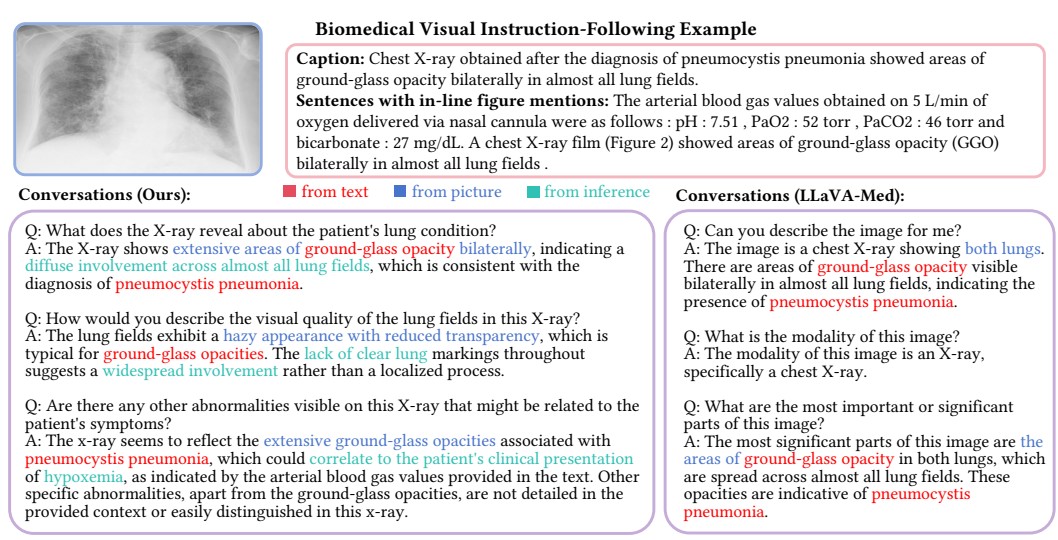

Figure 4: Case study on the generated instruction-following data.

**Generated instruction-following data.** We present case studies of the instructional data produced by BioMed-VITAL and the baseline LLaVA-Med in Figure 4, where the instruction data generated by both the input image and captions are presented in the left and right panels, respectively.

Regarding instruction generation, BioMed-VITAL generates instructions/questions that are closely related to clinical contexts and delves deeply to prompt in-depth discussions. For instance, we noticed that instructions of LLaVA-Med tend to be basic, such as "What is the modality of this image?", which lack targeted in-depth exploration and fail to meet the requirements for in-depth biomedical understanding and clinical relevance. In comparison, the question "What does the X-ray reveal about the patient's lung condition?" from BioMed-VITAL clarifies the specific organ and encourages a deeper understanding of the image by correlating observable features.

In terms of response generation, we differentiate the sources of the generated answers using different colors: red highlights indicate information derived from the input caption, blue highlights correspond to information based on the image, and green highlights information deduced by the model through reasoning and inference. It shows that BioMed-VITAL can capture more accurate and comprehensive key information from texts and images and provide richer inference, potentially supporting complex medical reasoning and diagnostic tasks. Additional cases are in Appendix F Figure 10.

**Open-ended biomedical visual chat.** Appendix F Figure 11 presents a case study comparing the open-ended visual chat responses generated by our model BioMed-VITAL and the baseline LLaVA-Med model. While the baseline model provides detailed information about brain structures and functions, it fails to offer specific insights directly related to the given question. In contrast, BioMed-VITAL demonstrates superior performance by generating responses that directly address the question based on the provided imaging data. Our model identifies and describes different pathological states, such as control and depression, and interprets the implications of color variations in the image, indicating higher or lower uptake. This showcases a deeper understanding of the imaging data and highlights our model's ability to interact effectively in the given context. Moreover, the strong connection between the image and the generated text, along with the logical flow present in our model's answers, further emphasizes the robust capabilities of our trained models.

**Benchmark visual question answering.** Appendix F Figure 12presents case studies on benchmarks of BioMed-VITAL and LLaVA-Med before fine-tuning. In the examples from the VQA-RAD and SLAKE datasets, BioMed-VITAL provides straightforward and accurate responses by clearly stating "Yes" or "No" at the beginning of its answer and identifying critical features that were overlooked by the compared model. This improves overall accuracy, demonstrating its ability to focus on the most relevant information and provide concise, accurate answers. Furthermore, BioMed-VITAL demonstrates a high level of interpretability, which is exemplified in the context of the PathVQA dataset. As shown in the examples, the responses from BioMed-VITAL go beyond providing simple, direct answers. Instead, it offers comprehensive explanations that include relevant features and insights drawn from the pathological images, serving as the basis for its conclusions. By incorporating this interpretability, BioMed-VITAL not only answers the questions accurately but also provides a clear rationale for its decisions, enhancing the depth and quality of the analysis.

# 5   Conclusion and Discussion

In this work, we introduce BioMed-VITAL, a data-centric framework for biomedical visual instruction tuning that effectively aligns with clinician preferences. By incorporating clinician expertise into both the data generation and selection processes, BioMed-VITAL produces high-quality datasets that significantly enhance the performance of visual instruction tuning models in the biomedical domain. The data generation stage employs a diverse set of clinician-selected demonstrations to guide GPT-4V in generating instructional data that closely reflects the nuanced expectations of medical professionals. The data selection stage involves training a separate selection model that distills clinician preferences to select the most relevant and informative data, which shows superior alignment with human preference compared to GPT-4. The instruction-tuned model trained using the BioMed-VITAL framework demonstrates remarkable performance in downstream tasks. Our datasets and models are available on the Hugging Face repository `https://huggingface.co/BioMed-VITAL`.

**Limitation and discussion.** The images and texts we used for curating instruction-following datasets are taken from the PMC-15M, which includes image-text pairs from the five most common imaging modalities: Chest X-ray, MRI, Histology, Gross pathology, and CT. However, despite the variety, the dataset is not evenly distributed across modalities, with a larger number of radiology images compared to gross pathology. Such imbalance in modalities may introduce potential bias in the model's instruction tuning. Another limitation is the use of majority voting for aggregating annotations from different annotators. There is potential for more advanced conflict-handling mechanisms, such as penalizing high-variability samples to improve model confidence. Regarding the extension of this work, investigating the generalizability of BioMed-VITAL to other specialized domains is a valuable direction. While our focus is on biomedical vision-language models, the core techniques in BioMed-VITAL are designed to be adaptable, allowing researchers and practitioners in different fields to create high-quality instruction training datasets tailored to their specific needs, especially when they want to effectively distill expert preferences with only a few human expert annotations.

# 6   Acknowledgments

This research was partially supported by the National Institute Of Diabetes And Digestive And Kidney Diseases of the National Institutes of Health under Award Number K25DK135913, the Emory Global Diabetes Center of the Woodruff Sciences Center, Emory University, and the Microsoft Accelerating Foundation Models Research (AFMR) grant program.

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

# A  Prompt for Instructional Data Generation

| The Prompt for Generating Instruction-Following Data with GPT-4V |
| --- |
| messages = [ {"role": "system", "content": """You are an AI assistant specialized in biomedical topics. You are provided with a figure image from a biomedical research paper. In some cases, you may have additional text (Figure Context) that mentions the image. Your task is to facilitate a dialogue where a person (User) seeks information about the image, and you (Assistant) provide insightful responses. During this interaction, the conversation should evolve as if both the User and Assistant are observing the image together. It is essential to thoroughly consider and reference the accompanying textual information (Figure Caption and Figure Context) and visual information to ensure a rich and informative exchange that highlights the significance of the visual details present.

Please meticulously extract all possible visual details from the image, and when generating instructions and responses, ensure to integrate and consider the provided supplementary textual information. It is crucial to highlight the connections and correlations between the textual content and the visual elements within the picture to capture the full context.

Below are the requirements for generating the instructions and responses in the conversation:
- Focus on visual aspects of the image that can be inferred without the text information, and extract as much key detailed information from the image as possible.
- Ensure that instructions are diverse and cover a range of visual aspects of the image.
- The conversation should encompass a minimum of 4-5 exchanges of instructions and responses. You may adjust the number of rounds based on the provided image and text. For content with substantial information, employing additional instructions and responses may be more appropriate to ensure thorough discussion and understanding.
- When the provided textual information is relevant to the instruction, try to respond using the expertise and specialized terminology contained within the text, rather than with vague, non-specialized descriptions. """} ]

for sample in fewshot_samples:
    messages.append({"role": "user", "content": sample['context']})
    messages.append({"role": "assistant", "content": sample['response']})

messages.append({"role": "user", "content": [image, text]}) |

Figure 5: The prompt we used for generating instruction-following data with GPT-4V. At each call, a set of examples sampled from diverse clinician-selected samples is included in the prompt as few-shot demonstrations, in which each example includes the 'context' and 'response'. The message concludes with an image and text as the query for the instruction-following generation.

The detailed prompt for instruction-following data generation with GPT-4V is shown in Figure 5.

# B  Clinician Preference Annotation and Model Preference Generation

The detailed prompt for generating instructions and their two candidate responses for clinician preference annotation is shown in Figure 6; the annotation of clinician preference is shown in Figure 7. Specifically, clinicians are asked to compare two answer candidates for each given instruction and choose the better response. They can select both if two responses are equally good or deselect both to drop the instruction. The figure contains real examples of clinician annotations.

The prompt for model-based preference generation is shown in Figure 8.

# C  Model Ablation Results on VQA Benchmarks

We have conducted additional experiments with recent language-vision models, including LLaVA-OneVision [1] and InternVL-1.5 [2], trained on our generated instruction data and evaluated on three VQA benchmarks. The results are summarized in Table 5. In addition, we conducted experiments using models of various sizes (7B and 13B parameters). The results are presented in Table 6.



**The Prompt for Generating Two Candidate Responses for Annotation**

messages = [ {"role": "system", "content": """ You are an AI assistant with expertise in biomedicine. You will be presented with an image from a biomedical research paper. Sometimes, additional textual information (figure captions and inline mentions) related to the image may be provided. Please meticulously extract key information from the visual context of the image and integrate it with the provided textual context, then generate conversations composed of questions from Users and answers from an AI assistant. It is crucial to highlight the connections and correlations between the textual content and the visual elements to capture complete and valuable information.

For every generated question, you are required to craft two separate candidate answers (Answer 1 and Answer 2). Each answer should offer a unique explanation or perspective based on the visual and textual contexts. This dual-answer format is intended to present alternative insights and to help determine which response might be more informative or appropriate.

Here are the guidelines for the dialogue:
- Do not use specific facts, terms, abbreviations, dates, numbers, or names from the supplementary text to ensure the focus remains on the image itself.
- Create diverse questions that explore different visual aspects of the image.
- Provide a minimum of 3-4 rounds (each round includes one question and two answers) of Q&A, with each question followed by two distinct answers.
- Respond thoughtfully, without overconfidence, and refrain from offering medical advice or diagnoses. Encourage seeking professional medical consultation.
- Initiate the conversation with a question from the User, not with an answer from the Assistant.
- Extract and utilize as much critical visual information from the image as possible, complemented by the additional text provided."""} ]

messages.append({"role": "user", "content": [image, text]})



Figure 6: The prompt we used for generating two candidate responses for preference annotation with GPT-4V.

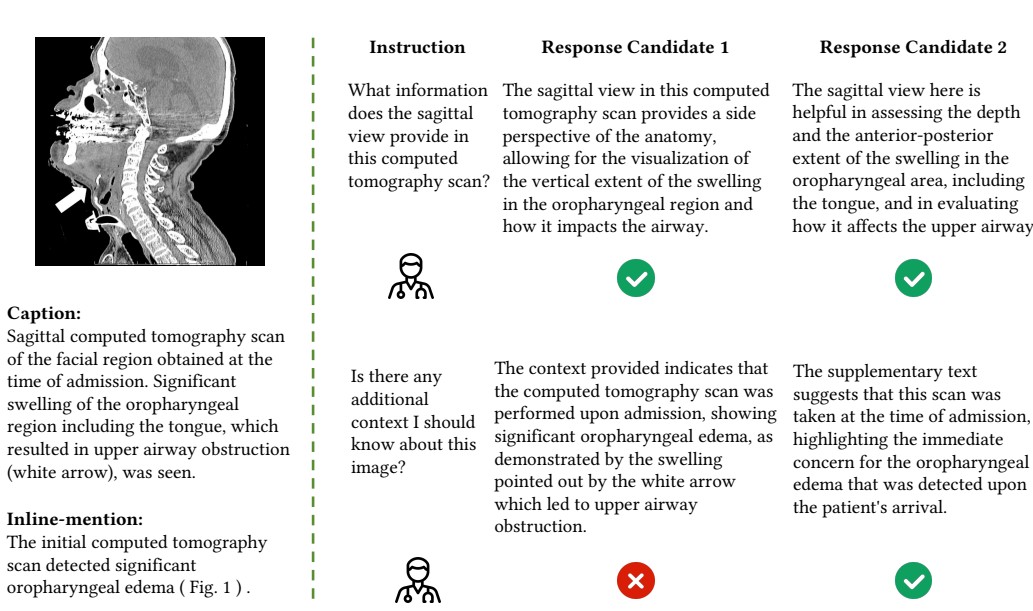

Figure 7: Clinician preference annotation.

## D   Details of Compared Methods on VQA Benchmarks

We include the details of each compared method in Table 4 for biomedical VQA benchmarks.

- **MMQ (Multi-Modal Question Answering)** [9] focuses on enhancing medical VQA using meta-learning to manage data quality and improve model robustness for better accuracy.

Figure 8: The prompt for model-based preference generation based on the clinician-curated criteria.

Table 5: Performance comparison across different models and training approaches.

| Language-Vision Model | Medical Instruct-Tuning | VQA-RAD | | SLAKE | | PathVQA | |
|---|---|---|---|---|---|---|---|
| | | Open | Closed | Open | Closed | Open | Closed |
| LLaVA | No | 50.00 | 65.07 | 78.18 | 63.22 | 7.74 | 63.20 |
| LLaVA | Yes (w/ LLaVA-Med Dataset) | 61.52 | 84.19 | 83.08 | 85.34 | 37.95 | 91.21 |
| LLaVA | Yes (w/ Ours) | 63.46 | 84.71 | 85.41 | 87.26 | 38.96 | 92.39 |
| LLaVA-OneVision | No | 51.26 | 69.49 | 76.45 | 63.46 | 8.24 | 67.50 |
| LLaVA-OneVision | Yes (w/ LLaVA-Med Dataset) | 62.37 | 78.68 | 83.54 | 86.29 | 37.44 | 92.06 |
| LLaVA-OneVision | Yes (w/ Ours) | 62.43 | 83.82 | 85.59 | 87.02 | 39.25 | 92.18 |
| InternVL-1.5 | No | 49.31 | 60.29 | 76.63 | 62.74 | 9.37 | 63.25 |
| InternVL-1.5 | Yes (w/ LLaVA-Med Dataset) | 61.22 | 81.25 | 84.24 | 81.97 | 37.14 | 91.53 |
| InternVL-1.5 | Yes (w/ Ours) | 62.38 | 84.92 | 86.75 | 83.17 | 39.20 | 92.65 |

Table 6: Performance comparison across different model sizes.

| Method | Size | VQA-RAD | | SLAKE | | PathVQA | |
|---|---|---|---|---|---|---|---|
| | | Open | Closed | Open | Closed | Open | Closed |
| LLaVA | 7B | 50.00 | 65.07 | 78.18 | 63.22 | 7.74 | 63.20 |
| LLaVA | 13B | 52.23 | 63.23 | 76.59 | 64.42 | 8.82 | 66.32 |
| LLaVA-Med | 7B | 61.52 | 84.19 | 83.08 | 85.34 | 37.95 | 91.21 |
| LLaVA-Med | 13B | 64.58 | 77.94 | 84.97 | 85.58 | 38.82 | 92.39 |
| BioMed-VITAL | 7B | 63.46 | 84.71 | 85.41 | 87.26 | 38.96 | 92.39 |
| BioMed-VITAL | 13B | 64.88 | 84.55 | 87.82 | 86.54 | 39.71 | 91.41 |

- **Prefix T. Medical LM** [40] leverages pre-trained language models with visual prefixes, excelling on SLAKE and PathVQA.
- **PubMedCLIP** [10] fine-tune the CLIP on PubMed data, demonstrating the potential of domain-specific adaptations for substantial performance gains.
- **BiomedCLIP** [46] uses a large-scale biomedical dataset for contrastive pretraining, achieving notable performance for medical vision-language tasks.
- **M2I2 (Multi-Modal Integration and Interaction)** [21] combines masked image modeling and contrastive learning, leading to a high 88.00% accuracy on PathVQA.
- **MUMC (Multi-Modal Unified Model for Clinical Tasks)** [20] integrates both unimodal and multimodal contrastive losses, achieving high results on VQA-RAD.
- **M3AE (Multi-Modal Masked Autoencoder)** [5] employs multi-modal masked autoencoders in a self-supervised learning setup to enhance cross-modal performance.
- **CoQAH (Chain of Question Answering for Human-written Question** [16] utilizes iterative QA interactions between a large language model and a VQA model to answer complex visual questions, achieving high accuracy without fine-tuning.
- **PMC-CLIP** [22] pre-trains a vision-language model on a large-scale biomedical dataset with 1.6M image-caption pairs to improve various medical visual tasks such as retrieval and classification.

## E  Prompt for Win Rate Evaluation on VQA Benchmarks

| **The Prompt for Reference-Guided Pairwise Win-Rate Evaluation on VQA Benchmarks** |
|---|
| messages = [ {"role": "system", "content": """ Please act as an impartial judge and evaluate the quality of the responses provided by two AI assistants to the user question displayed below. |
| Your evaluation should consider correctness and helpfulness. You will be given a reference answer, assistant A's answer, and assistant B's answer. Your job is to evaluate which assistant's answer is better. Begin your evaluation by comparing both assistants' answers with the reference answer. Identify and correct any mistakes. Avoid any position biases and ensure that the order in which the responses were presented does not influence your decision. |
| Do not allow the length of the responses to influence your evaluation. Do not favor certain names of the assistants. Be as objective as possible. |
| After providing your explanation, output your final verdict by strictly following this format: "[[A]]" if assistant A is better, "[[B]]" if assistant B is better, and "[[C]]" for a tie. You must begin with [[A]] or [[B]] or [[C]]. Assigning "[[C]]" should be a last resort, used only if you absolutely cannot discern any difference in the quality of the two responses. """} ] |
| |
| messages.append({"role": "user", "content": [image, question]}) |
| messages.append({"role": "user", "content": reference_answer}) |
| messages.append({"role": "user", "content": (answer_A, answer_B)}) |

Figure 9: The prompt for win-rate evaluation on VQA benchmarks.

The detailed prompt for win rate evaluation on VQA benchmarks is shown in Figure 9.

## F  Case Study

Case studies on the generated instruction-following data are presented in Figure 10. Case studies for the downstream task of open-ended visual chat and the downstream task of biomedical VQA benchmarks are presented in Figure 11 and Figure 12. Detailed analysis of these case studies can be found in Section 4.5.

## G  Key Information

This work focuses on introducing **an effective data-centric practice for creating and curating datasets** used for biomedical visual instruction tuning. The datasets produced from our proposed framework, named BioMed-VITAL, are released as by-products of the core contribution. Therefore, we only include relevant and key information as required.

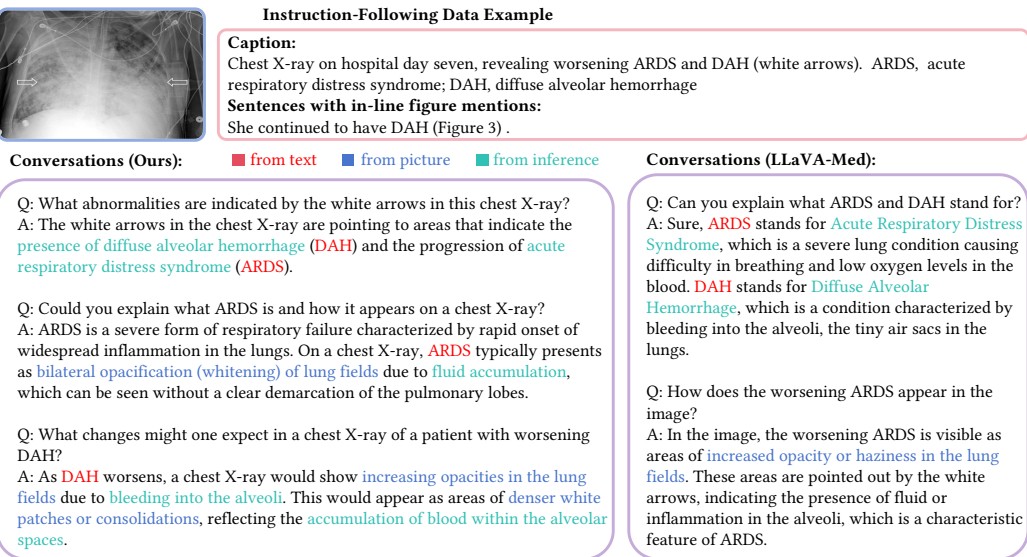

Figure 10: Additional case study for generated instruction-following data.

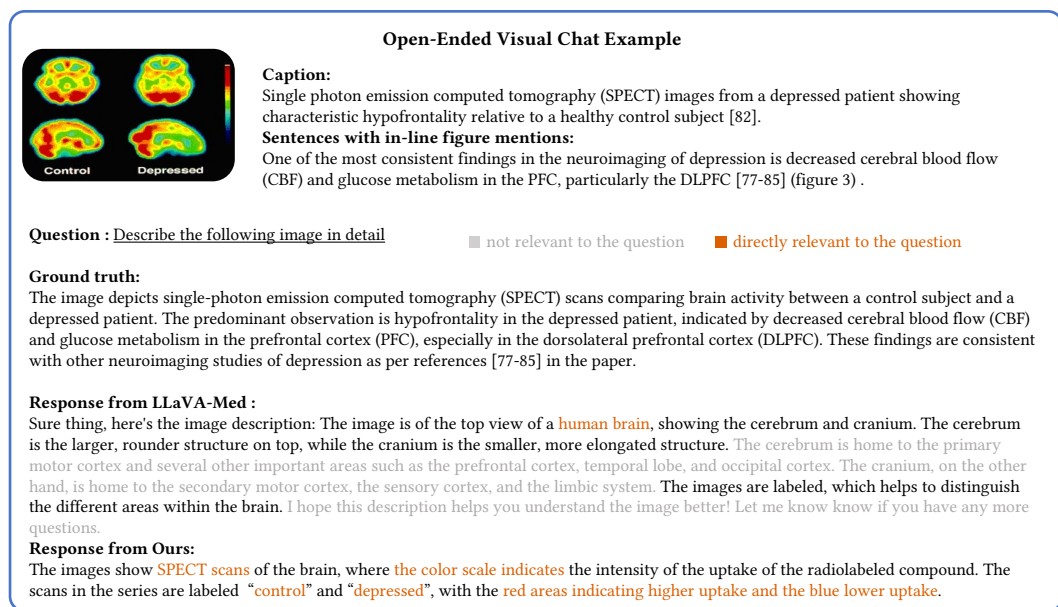

Figure 11: Case study for the downstream task of open-ended visual chat.

## G.1 Dataset Documentation

We release the instruction-following datasets curated from our framework, provided in `json` format. Each instructional data point contains the following fields:

- `id`: a unique identifier for the example;
- `image`: the image associated with the example;
- `domain`: the domain of the image, which includes `CXR`, `MRI`, `Histology`, `Gross`, and `CT`;
- `conversations`: a sequence of 4-5 rounds of instructions and responses related to the image.

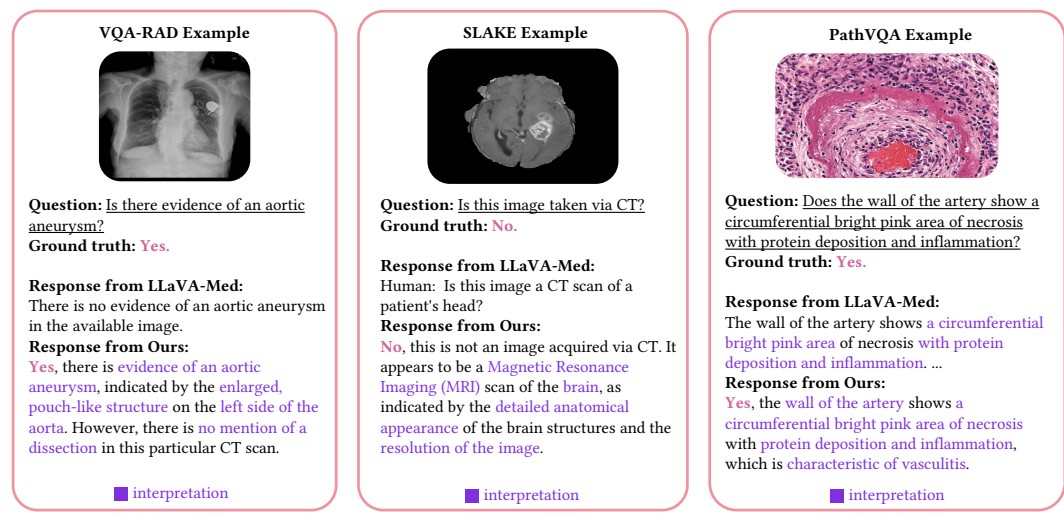

Figure 12: Case study for the downstream task of biomedical VQA benchmarks.

## G.2 Intended Uses

The datasets are intended for researchers in machine learning and language models, particularly in the field of health ML and related areas. It aims to facilitate the development and adaptation of large multimodal models to meet the real-world needs of clinicians. The proposed data-centric methods incorporate clinician preferences into the dataset curation process and can be applied to other specialized domains lacking annotated data for domain adaptation.

## G.3 Hosting and Maintenance Plan

The datasets and models are available on the Hugging Face repository `https://huggingface.co/BioMed-VITAL`. All datasets can be directly accessed and downloaded from this repository. The authors will be responsible for maintaining the datasets. We welcome contributions from external contributors to expand the dataset with additional medical imaging domains and enhance conversational annotations to support more complex interaction scenarios.

## G.4 Licensing

We distribute the curated instructional datasets under a standard CC-BY-4.0 license. Models trained using the dataset should not be used for non-research purposes. All the resources are also restricted to uses that comply with the license agreements of CLIP, LLaMA, LLaVA, and GPT-4.

## G.5 Author Statement

We, the authors, will bear all responsibility in case of violation of rights and confirmation of date license.

## G.6 Reproducibility

All necessary resources, including source code, model checkpoints, training configurations, and detailed instructions for replicating our results, are publicly accessible through our project website at `https://BioMed-VITAL.github.io`.

