## A  Prompt for Instructional Data Generation

The detailed prompt we used for instruction-following data generation with GPT-4V is shown in Figure 5.

| The Prompt for Generating Instruction-Following Data with GPT-4V |
|---|

```
messages = [ {"role": "system", "content": """You are an AI assistant specialized in biomedical topics. You
are provided with a figure image from a biomedical research paper. In some cases, you may have additional
text (Figure Context) that mentions the image. Your task is to facilitate a dialogue where a person (User) seeks
information about the image, and you (Assistant) provide insightful responses. During this interaction, the
conversation should evolve as if both the User and Assistant are observing the image together. It is essential to
thoroughly consider and reference the accompanying textual information (Figure Caption and Figure Context)
and visual information to ensure a rich and informative exchange that highlights the significance of the visual
details present.

Please meticulously extract all possible visual details from the image, and when generating instructions and
responses, ensure to integrate and consider the provided supplementary textual information. It is crucial to
highlight the connections and correlations between the textual content and the visual elements within the
picture to capture the full context.

Below are requirements for generating the instructions and responses in the conversation:
- Focus on visual aspects of the image that can be inferred without the text information, and extract as much
key detailed information from the image as possible..
- Ensure that instructions are diverse and cover a range of visual aspects of the image.
- The conversation should encompass a minimum of 4-5 exchanges of instructions and responses. You may
adjust the number of rounds based on the provided image and text. For content with substantial information,
employing additional instructions and responses may be more appropriate to ensure thorough discussion and
understanding.
-   When the provided textual information is relevant to the instruction, try to response using the expertise and
    specialized terminology contained within the text, rather than with vague, non-specialized descriptions."""} ]

for sample in fewshot_samples:
    messages.append({"role": "user", "content": sample['context']})
    messages.append({"role": "assistant", "content": sample['response']})

messages.append({"role": "user", "content": [image, text]})
```

Figure 5: The prompt we used for generating instruction-following data with GPT-4V. At each call, a set of examples sampled from diverse clinician-selected samples is included in the prompt as few-shot demonstrations, in which each example includes the 'context' and 'response'. The message concludes with an image and text as the query for the instruction-following generation.

## B  Prompt for Model-Based Preference Generation

The detailed prompt for model-based preference generation is shown in Figure 6.

## C  Prompt for Win Rate Evaluation on VQA Benchmarks

The detailed prompt for win rate evaluation on VQA benchmarks is shown in Figure 7.

## D  Details of Benchmark Datasets

The statistics of the three benchmark datasets are shown in Table 4.

- VQA-RAD [14] is a dataset containing 3,515 question-answer pairs created by medical professionals, along with 315 radiology images. Each image is linked to several questions, which are categorized into 11 types, including abnormality, attribute, modality, organ system, color, counting, object/condition presence, size, plane, positional reasoning, and others. The dataset features a balanced mix of closed-ended (yes/no) and open-ended (one-word or short phrase) answers.

---

**The Prompt for Model-Based Preference Generation Based on the Clinician-Curated Criteria**

messages = [ {"role": "system", "content": """ Assume that you are a medical expert with extensive experience in your field. Your task is to assess and score a set of question-and-answer pairs from an instruction following dataset designed for fine-tuning a medical large language model (LLM). You should give score for each Q-A pair as you are provided with multi-round conversations. You will be provided with images, their corresponding captions, in-text mentions, and the Q&A pairs. Your scoring, ranging from 0 to 10, will evaluate the following criteria:

- Scope and Relevance: How well does the question cover key aspects of the medical image and caption provided?
- Value for Fine-Tuning: Is the question formulated in a way that it will add value to the fine-tuning process of the medical LLM?
- Answer Alignment: Does the provided answer directly address the question posed?
- Accuracy: Is the information in the answer medically accurate and correct?
- Utility: How useful is the answer in a medical context? Does it provide actionable or insightful information?
- Image Content Recognition and Utilization: Does the response accurately identify the content depicted in the image and effectively incorporate this information into the answer to enhance comprehension or applicability in a medical context?

Please consider additional factors such as:
- Clarity: Are both the question and the answer clearly articulated and free of ambiguity?
- Detail and Depth: Do the answer's details contribute to a deeper understanding of the topic?
- Medical Precision: How precisely do the question and answer reflect medical terminology and knowledge?

As you review each Q&A pair, Please first output a single line containing scores of each Q&A pairs, splited by blank. The first score is for first question and its corresponding answer, and so on. After that you can give some explanations, like what are the shortcomings of the current instructions. """} ]

messages.append({"role": "user", "content": [image, text]})
messages.append({"role": "user", "content": (instruction, response)})

---

Figure 6: The prompt for model-based preference generation based on the clinician-curated criteria.

---

**The Prompt for Reference-Guided Pairwise Win-Rate Evaluation on VQA Benchmarks**

messages = [ {"role": "system", "content": """ Please act as an impartial judge and evaluate the quality of the responses provided by two AI assistants to the user question displayed below.
Your evaluation should consider correctness and helpfulness. You will be given a reference answer, assistant A's answer, and assistant B's answer. Your job is to evaluate which assistant's answer is better. Begin your evaluation by comparing both assistants' answers with the reference answer. Identify and correct any mistakes. Avoid any position biases and ensure that the order in which the responses were presented does not influence your decision.
Do not allow the length of the responses to influence your evaluation. Do not favor certain names of the assistants. Be as objective as possible.
After providing your explanation, output your final verdict by strictly following this format: "[[A]]" if assistant A is better, "[[B]]" if assistant B is better, and "[[C]]" for a tie. You must begin with [[A]] or [[B]] or [[C]]. Assigning "[[C]]" should be a last resort, used only if you absolutely cannot discern any difference in the quality of the two responses. """} ]

messages.append({"role": "user", "content": [image, question]})
messages.append({"role": "user", "content": reference_answer})
messages.append({"role": "user", "content": (answer_A, answer_B)})

---

Figure 7: The prompt for win-rate evaluation on VQA benchmarks.

- SLAKE [19] is a comprehensive medical visual question-answering dataset with knowledge-enhancement features. It contains radiology images and diverse question-answer pairs annotated by experienced physicians. The dataset incorporates external medical knowledge through a provided medical knowledge graph, and the images are supplemented with rich visual annotations, including semantic segmentation masks and object detection bounding boxes. SLAKE covers a wide range of modalities and human body parts, such as the brain, neck, chest, abdomen, and pelvic cavity. We adopt only the English subset of SLAKE in our experiments.

Table 4: Statistics of the benchmark datasets for downstream evaluation on biomedical VQA.

| Dataset | VQA-RAD | | SLAKE | | | PathVQA | | |
|---|---|---|---|---|---|---|---|---|
| | Train | Test | Train | Val | Test | Train | Val | Test |
| # Images | 313 | 203 | 450 | 96 | 96 | 2,599 | 858 | 858 |
| # QA Pairs | 1,797 | 451 | 4,919 | 1,053 | 1,061 | 19,755 | 6,279 | 6,761 |
| # Open | 770 | 179 | 2,976 | 631 | 645 | 9,949 | 3,144 | 3,370 |
| # Closed | 1,027 | 272 | 1,943 | 422 | 416 | 9,806 | 3,135 | 3,391 |

- PathVQA [10] focuses on pathology images. Each image is associated with multiple questions that cover various aspects, such as location, shape, color, and appearance. The questions in PathVQA include open-ended questions (e.g., why, what, how, where) and closed-ended questions.

# E    Details of Compared Methods on Benchmarks

Below, we include the design details of each compared method for biomedical VQA benchmarks as shown in Table 3.

- **MMQ (Multi-Modal Question Answering)** [6] focuses on enhancing medical VQA using meta-learning to manage data quality and improve model robustness for better accuracy.
- **Prefix T. Medical LM** [35] Prefix T. Medical LM leverages pre-trained language models with visual prefixes, excelling on SLAKE and PathVQA.
- **PubMedCLIP** [7] fine-tunes the CLIP on PubMed data, demonstrating the potential of domain-specific adaptations for substantial performance gains.
- **BiomedCLIP** [41] uses a large-scale biomedical dataset for contrastive pretraining, achieving notable performance for medical vision-language tasks.
- **M2I2 (Multi-Modal Integration and Interaction)** [17] combines masked image modeling and contrastive learning, leading to a high 88.00% accuracy on PathVQA.
- **MUMC (Multi-Modal Unified Model for Clinical Tasks)** [16] integrates both unimodal and multimodal contrastive losses, , achieving high results on VQA-RAD.
- **M3AE (Multi-Modal Masked Autoencoder)** [3] employs multi-modal masked autoencoders in a self-supervised learning setup to enhance cross-modal performance.
- **CoQAH (Collaborative Question Answering in Healthcare)** [13] utilizes iterative QA interactions between a large language model and a VQA model to answer complex visual questions, achieving high accuracy without fine-tuning.
- **PMC-CLIP** [18] pre-trains a vision-language model on a large-scale biomedical dataset with 1.6M image-caption pairs to improve various medical visual tasks such as retrieval and classification.

# F    Additional Case Study

Additional case studies on the generated instruction-following data and downstream evaluations of open-ended visual chat and biomedical VQA benchmark are presented in Figure 8, Figure 9, and Figure 10 respectively. Detailed analysis of the case studies can be found in Section 4.5.

# G    Limitation

The images and texts used in this work for curating instruction-following datasets and demonstrating the data-centric framework are taken from the PMC-15M, which includes image-text pairs from the five most common imaging modalities: Chest X-ray, MRI, Histology, Gross pathology, and CT. However, despite the variety indicated in LLaVA-Med [15], the dataset is not evenly distributed across modalities, with a larger number of radiology images compared to gross pathology. Such imbalance in modalities may introduce potential bias in the model's instruction tuning. Although the proposed method can be applied to other resources, we did not attempt it on additional datasets due to limitations in computation resources. Another limitation is that even after the data selection

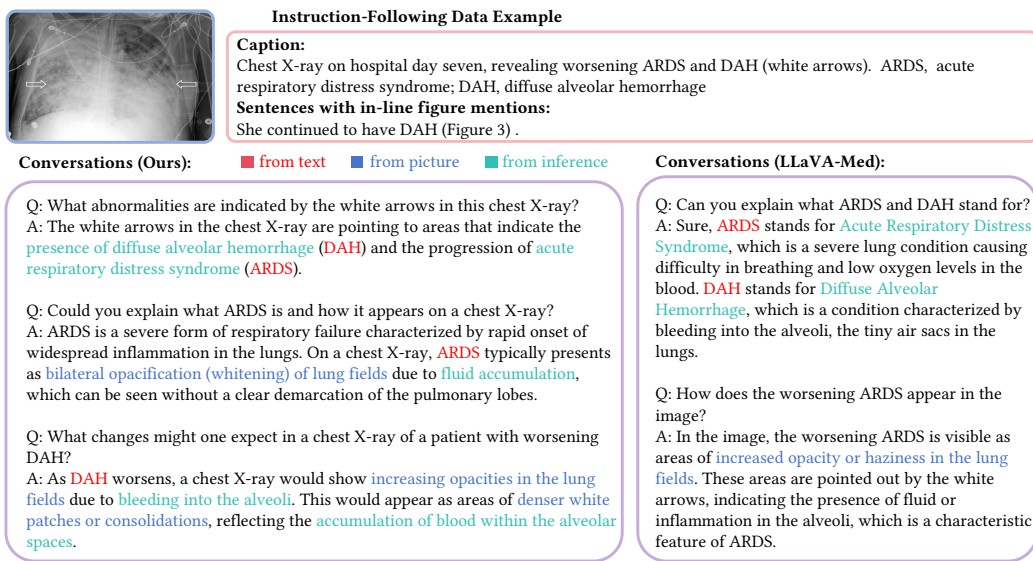

Figure 8: Additional case study for generated instruction-following data.

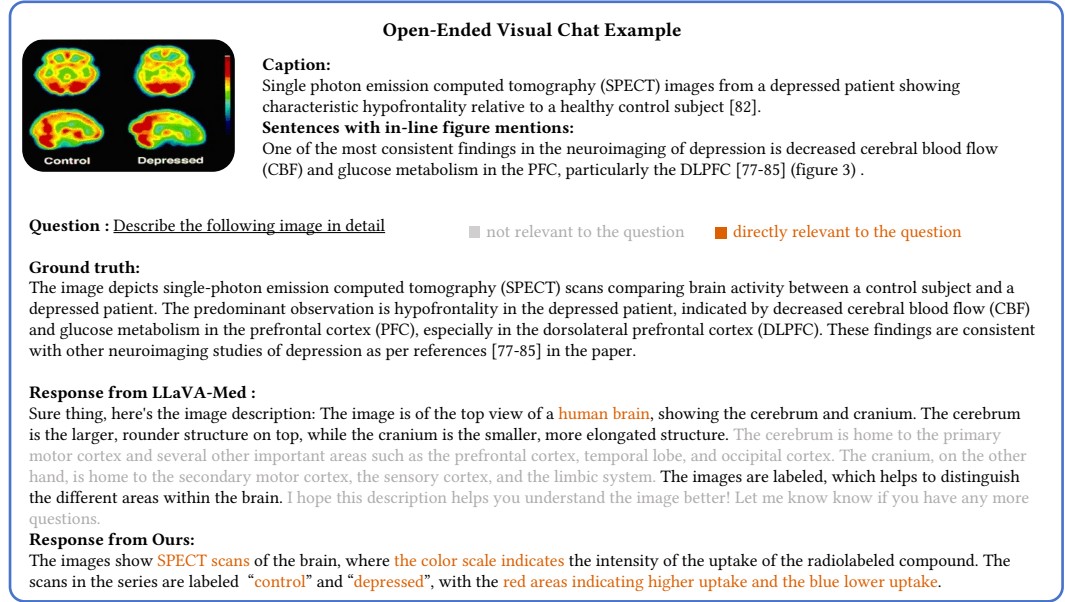

Figure 9: Case study for the downstream task of open-ended visual chat.

process, noise and low-quality instruction data may remain. The issue of hallucination arising from the data generation might not be fully addressed and is worth future efforts to improve.

## H  Clinician Preference Annotation

The annotation of clinician preference is shown in Figure 11. Specifically, clinicians are asked to compare two answer candidates for each given instruction and choose the better response. They can select both if two responses are equally good or deselect both to drop the instruction. The figure contains real examples of clinician annotations.

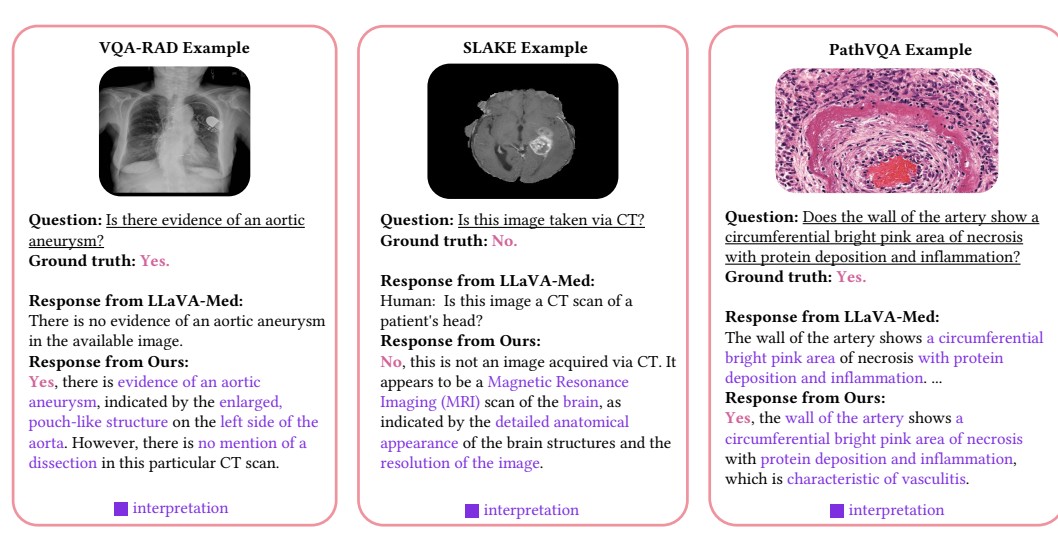

Figure 10: Case study for the downstream task of biomedical VQA benchmarks.

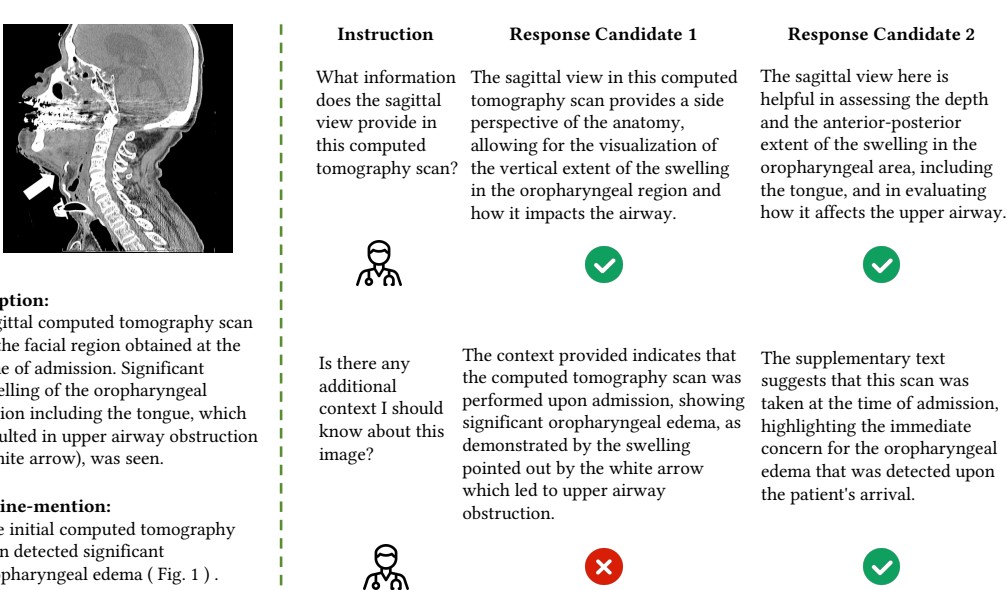

Figure 11: Clinician preference annotation.

# I Key Information

This work focuses on introducing **an effective data-centric practice for creating and curating datasets** used for biomedical visual instruction tuning. The datasets produced from our proposed framework, named BioMed-VITAL, are released as by-products of the core contribution. Therefore, we only include relevant and key information as required.

## I.1 Dataset Documentation

We release the instruction-following datasets curated from our framework, provided in `json` format. Each instructional data point contains the following fields:

- `id`: a unique identifier for the example;
- `image`: the image associated with the example;
- `domain`: the domain of the image, which includes `CXR`, `MRI`, `Histology`, `Gross`, and `CT`;

- `conversations`: a sequence of 4-5 rounds of instructions and responses related to the image.

## I.2  Intended Uses

The datasets are intended for researchers in machine learning and language models, particularly in the field of health ML and related areas. It aims to facilitate the development and adaptation of large multimodal models to meet the real-world needs of clinicians. The proposed data-centric methods incorporate clinician preferences into the dataset curation process and can be applied to other specialized domains lacking annotated data for domain adaptation.

## I.3  Hosting and Maintenance Plan

The datasets and models are hosted and version-tracked via Hugging Face. They will be permanently available under the repository `https://huggingface.co/datasets/mao1207/BioMed-VITAL-instructions`. All datasets can be directly accessed and downloaded from this repository. We plan to include expanding the dataset with additional medical imaging domains and enhancing conversational annotations to support more complex interaction scenarios. We encourage participation from external contributors. The authors will be responsible for maintaining the datasets.

## I.4  Licensing

We distribute the curated instructional datasets under a standard CC-BY-4.0 license. Models trained using the dataset should not be used for non-research purposes. All the resources are also restricted to uses that comply with the license agreements of CLIP, LLaMA, LLaVA, and GPT-4.

## I.5  Author Statement

We, the authors, will bear all responsibility in case of violation of rights and confirmation of date license.

## I.6  Reproducibility

All the prompts, instructions, and model checkpoints for reproducing the results can be found in the GitHub repository `https://github.com/mao1207/BioMed-VITAL`.