# OpenReview forum: "Biomedical Visual Instruction Tuning with Clinician Preference Alignment"
_NeurIPS.cc/2024/Datasets_and_Benchmarks_Track — NeurIPS 2024 Track Datasets and Benchmarks Poster_

### Official Review · Reviewer_s4zq · 2024-06-27
**Good work but many questions**

**Rating:** 7
**Confidence:** 4
**Correctness:** Mainly yes, the evaluaiton with GPT-4…
**Clarity:** Mainly yes, see questions and issues …

**Review:**

The alignment of neural networks with human perception is one of the most important topics of our time in my opinion. The proposed method and framework look good and straight forward. It is on the simpler side but this is not issue as long as it works. The effort of providing documentation, evaluations and comparison is visible and makes the work good to read. The results look very promising. This is not a completly new idea ,collecting examples and finetune on them, but the details are explained well and seem to be important. During reading I noticed several questions and issues, listed below.

- You state "From these clusters, we uniformly select a subset S with total M samples that have relatively complex captions and inline mentions." Please explain more since the selection process is not described at all.
- About the next sentence: How where the metioned instructions and candidate responses generated?
- How do you handle both selection or none selected in stage 2? You state only one correct item is selected in line 133+134
- How do you handle disagreement between annotators? Are you even considering this case?
- Is the loss in equation (1) not inherintly flawed? Example: three examples A,B;C and their scores R_A, R_B, R_C with R_A > R_B > R_C when comparing B with A; A is the target, if B is compared, B is the target, Depending on the comparison the classification is different. I see why it still might work but I would assume training direclty for a ranking would be leading to better results. What is your justification for this approach?
- About equation (3) and table (1) when is the weight of 400 added, I would assume not to all samples but you write it that way.
- Is the results of the model GPT-4V in figure 2 the same model which was used to calculate R_model? If yes have you trained training your selection model with only R_model or only R_human results as comparison. The results look promising but with reference their are difficult to judge.
- Can you explain the dips in Figure 2 in the top k rank? Is this just random performance. What about the peak around 30%?
- Limitations should be part of the paper and not be in the appendix
- Evaluation 4.3 Do you have any rating of humans? It could be that you just learned better style of writing which GPT-4V prefers as it was also included in the training sample generation.
- What does the number in Table 2 mean. If every question has a score of 1 to 10 I would assume you report the mean score which is not possible for scores of 69 and so on. Then I thought you just added them up but them the individual parts should be summed the same as the overall one.
- Typo: LLava-med is not from NeurIPS 2024 (these are still under review :D)
- More information about the VQA benchmarks are desirable, I had to check other sources to understand metrics and concepts. Also introducing this information only the supplement makes the publication more difficult to understand.
- The win-rate perfomance evaluation might have the same GPT-4V issue as described above.
- Why are not the same questions used for both models in the case studies?
- Why are the case studies not part of the main paper? How where these selected? They look very good but if it is just one cherry picked results I'm unsure how reliable the evaluation is. Do you have evaluations of humans for the results?


To summarize, I have many questions, the evaluation with GPT-4V seems not to be very reliable, the disagreement between annotators seems not be considered (while beeing very important for alignment), a lot of information is in the supplement which should be in the main paper (please check if you can restructure to be more focused).

Overall, I think this work is promising but has too many edges to be accepted currently.

**Strengths:**

See above under review.

**Additional Feedback:**

See the review above.

**Documentation:**

Instructions how to get the images to the annotations would be great. Based on quick check it seems like they have to be manually collected from the original source.

This page: https://huggingface.co/datasets/mao1207/BioMed-VITAL-instructions
Shows an error: The full dataset viewer is not available (click to read why). Only showing a preview of the rows.
Without digging into it, it would be great if you coud resolve this.

**Ethics:**

No concerns

**Limitations:**

The limitations are not discussed in the main paper but only in the appendix. The main issues should be discussed in the main paper. Moreover the issue of different preferences of clinicians is not evaluated at all. I worked with medical stuff before and it is common case that they disagree. This disagremment seems not to be considered at all.

**Opportunities For Improvement:**

Please the questions and suggestions above under review.

**Relation To Prior Work:**

It seems to be sufficiently differnt from previous work

**Summary And Contributions:**

The paper is about aligning the output of LLMs more closely to the preferences of clinicians. They provide a framework for doing so, provide the created data and show potential improvements based on this framework and data.

---

> ### Author Rebuttal · Authors · 2024-08-17
>
> Thank you for your thorough and informative review of our work! Please find our clarifications and additional results below.
>
> > **Q1: Explain the detailed process of diverse few-shot demonstration selection**
>
> **A**: Our approach uses K-means clustering to increase the diversity of few-shot demonstrations in the instruction data generation prompt. Unlike LLaVA-Med, which only annotated six instruction-response examples and constantly used these six examples for each generation, we aim to improve the generated instruction data’s diversity by providing different demonstrations at each time of generating one data. In particular, each demonstration of a batch is sampled from different clusters grouped by embeddings. We perform K-means clustering with M=60 clusters, then select a subset S of 300 samples from these clusters with long captions and inline mentions. These selected samples are annotated by clinicians. The resulting questions, paired with preferred answers, serve as our few-shot demonstration pool.
>
> At each time of generating one data, we randomly select 2 samples for each of the 5 image modalities from the annotated pool and append them to the prompt (Ln. 118-119). This method ensures a wide range of medical scenarios are represented while reducing the burden of clinician annotation. The detailed prompt used for instruction data generation is illustrated in Appendix A, Figure 5. By employing this approach, we aim to improve the quality and diversity of generated instruction-response pairs, potentially enhancing our model's performance across various medical contexts. We apologize for not clearly illustrating this detail initially and will make sure to include them in our final version.
>
> > **Q2: How were the instructions and candidate responses generated?**
>
> **A**: The instructions and two candidate responses to each instruction (question) are generated with the following prompt:
> ```
> You are an AI assistant with expertise in biomedicine. You will be presented with an image from a biomedical research paper. Sometimes, additional textual information (figure captions and inline mentions) related to the image may be provided. Please meticulously extract key information from the visual context of the image and integrate it with the provided textual context, then generate conversations composed of questions from Users and answers from an AI assistant. It is crucial to highlight the connections and correlations between the textual content and the visual elements to capture complete and valuable information.
>
> For every generated question, you are required to craft two separate candidate answers (Answer 1 and Answer 2). Each answer should offer a unique explanation or perspective based on the visual and textual contexts. This dual-answer format is intended to present alternative insights and to help determine which response might be more informative or appropriate.
>
> Here are the guidelines for the dialogue:
> - Do not use specific facts, terms, abbreviations, dates, numbers, or names from the supplementary text to ensure the focus remains on the image itself.
> - Create diverse questions that explore different visual aspects of the image.
> - Provide a minimum of 3-4 rounds (each round includes one question and two answers) of Q&A, with each question followed by two distinct answers.
> - Respond thoughtfully, without overconfidence, and refrain from offering medical advice or diagnoses. Encourage seeking professional medical consultation.
> - Initiate the conversation with a question from the User, not with an answer from the Assistant.
> - Extract and utilize as much critical visual information from the image as possible, complemented by the additional text provided.
> ```
> We iterated on the generation prompt with clinicians and incorporated their review feedback to ensure the quality of the generation. We appreciate the question and will include the details in our final version!
>
> > **Q3: How to handle both selected and none selected pairs in Stage 2**
>
> **A**: Thank you for your question! When working with the clinician annotated data, we combined all samples with selected (label = 1) and unselected (label = 0) to form pairs for training data. This was done to increase the number of human preferences. Our assumption was that a sample with label 1 is always better than a sample with label 0. We chose this approach based on the observation that samples with an annotated label of 0 always have obvious flaws or are associated with unreasonable questions. In contrast, samples with an annotated label of 1 are usually at least correct or valuable.
>
> > **Q4: The variety/disagreement of clinicians’ preferences and how they are handled**
>
> **A**: Thank you for this insightful question! Three annotators, who are MDs and PhDs from Massachusetts General Hospital Radiology, participated in our process of annotating clinical data. They generated annotations by selecting the preferred response from two choices for 300 stratified samples. Out of these 300 annotations, clinicians agreed on the same decision in 241 cases. Disagreements mainly occurred when the two responses were of equal quality and difficult to differentiate, which was a relatively small portion of the annotations (59 out of 300). When all three annotators reached the same decision, we used that selection. For disagreements, we applied majority voting to make the final decision. As a measure of the consistency of human annotations, we calculated the Fleiss' kappa, resulting in a score of 0.736, which indicates relatively consistent agreement among the three clinician annotators.
>
> We recognize the limitations of using majority voting and acknowledge the potential for more advanced conflict-handling mechanisms, such as penalizing high-variability samples to improve model confidence, in the future. We will include this limitation discussion and insightful future improvements in our paper.

---

> ### Author Rebuttal · Authors · 2024-08-17
>
> > **Q5: Is the loss in equation (1) flawed? Justification for this approach vs. direct ranking?**
>
> **A**: We appreciate your insight. In our training, the data selection model learns to predict the likelihood of a sample being labeled as 1 within a sampled pair. Given our dataset and batch sizes, it is unlikely for a sample to be labeled as both 0 and 1 in the same batch. We conducted experiments comparing our loss function with ranking hinge loss, which is commonly used in learning-to-rank algorithms:
>
> $\mathcal{L}_Q = \max(0, 1 - y(f_i - f_j))$. when $R_i > R_j$ , $y = +1$; when $R_i < R_j$, $y = -1$.
>
> The results are presented below:
> Loss Function |  Mixture Strategy | ACC ↑ | AUC ↑ | MR ↓  | MAP ↑ |
> |-----|-----|-------|-------|-------|-------|
> |Ours | $w_{\mathcal{R}\_\text{human}}/w_{\mathcal{R}_\text{model}}$ = 100   |62.05 | 62.30 | 43.84 | 61.63 |
> |Ours | $w_{\mathcal{R}\_\text{human}}/w_{\mathcal{R}_\text{model}}$ = 200   | 60.91 | 61.23 | 44.37 | 59.55 |
> |Ours | $w_{\mathcal{R}\_\text{human}}/w_{\mathcal{R}_\text{model}}$ = 300   | 63.64| 63.12| 43.43 | 63.00 |
> |Ours| $w_{\mathcal{R}\_\text{human}}/w_{\mathcal{R}_\text{model}}$ = 400  | 66.72 | 66.32 | 41.83 | 64.47 |
> |Ours| $w_{\mathcal{R}\_\text{human}}/w_{\mathcal{R}_\text{model}}$ = 500  | 62.85 | 63.06 | 43.46 | 65.00 |
> |Raning Hinge Loss| $w_{\mathcal{R}\_\text{human}}/w_{\mathcal{R}_\text{model}}$ = 100 | 61.41 | 61.65 | 44.16 | 60.57 |
> |Raning Hinge Loss| $w_{\mathcal{R}\_\text{human}}/w_{\mathcal{R}_\text{model}}$ = 200 | 60.71 | 59.82 | 45.08 | 62.37 |
> |Raning Hinge Loss| $w_{\mathcal{R}\_\text{human}}/w_{\mathcal{R}_\text{model}}$ = 300 | 64.34 | 64.57 | 42.71 | 66.90 |
> |Raning Hinge Loss| $w_{\mathcal{R}\_\text{human}}/w_{\mathcal{R}_\text{model}}$ = 400 | 66.42 | 66.07 | 41.95 | 65.16 |
> |Raning Hinge Loss| $w_{\mathcal{R}\_\text{human}}/w_{\mathcal{R}_\text{model}}$ = 500 | 62.22 | 62.32 | 43.84 | 62.71 |
>
> The results demonstrate that our loss function performs competitively compared to the ranking hinge loss across various mixture strategies. Both loss functions show similar trends, with performance peaking around the 300-400 range for the mixture strategy. Despite its simpler formulation, our loss function proves to be robust and effective for data selection. We completely agree that improving the mixture strategy or exploring potentially improved learning objectives can further enhance the selection model’s performance for future work.
>
> > **Q6: How is the sample weight applied to Equation (3) and Table 1?**
>
> **A**: We appreciate your question and apologize for the confusion in the presentation of Equation (3) and Table 1. The sample weight serves as an expansion factor for human annotation compared to model-based preference.
>
> In Equation (3), the application of sample weight should be represented as:
>
> $\mathcal{L}\_Q=-w_{(i,j)} \left(z_i \log \sigma\left(f(x_i)\right)+z_j \log \sigma\left(f(x_j)\right)\right)$
>
> $w_{(i,j)}$ represents the adjustable expansion factor for samples from human annotation. It takes values from the set {1, 50, …, 400, …}, with the assumption that $w_{(i,j)}=1$ for pairs from model-based annotation. We will ensure that the notations in Equation 3 and Table 1 accurately reflect this in our final version.
>
> > **Q7: Ablation of training data selection model with only R_model or R_human.**
>
> **A**: Thanks for your insightful suggestion for this ablation study! We conducted experiments to train the data selection model using only human-generated preference $\mathcal{R}\_\text{human}$ or only model-generated ratings $\mathcal{R}\_\text{model}$.
>
> The results are summarized  in the following table:
> |  Mixture Strategy | ACC ↑ | AUC ↑ | MR ↓  | MAP ↑ |
> |-----|-------|-------|-------|-------|
> | only $\mathcal{R}_\text{human}$ | 55.89 | 55.99 | 46.91 | 56.21 |
> | only $\mathcal{R}_\text{model}$ | 54.76 | 54.64 | 47.67 | 55.25 |
> | mix, $w_{\mathcal{R}\_\text{human}} / w_{\mathcal{R}_\text{model}}$ = 5   | 58.63 | 58.22 | 45.87 | 62.29 |
> | mix, $w_{\mathcal{R}\_\text{human}} / w_{\mathcal{R}_\text{model}}$ = 10  | 59.38 | 59.14 | 45.39 | 59.20 |
> | mix, $w_{\mathcal{R}\_\text{human}} / w_{\mathcal{R}_\text{model}}$ = 100 | 62.05 | 62.30 | 43.84 | 61.63 |
> | mix, $w_{\mathcal{R}\_\text{human}} / w_{\mathcal{R}_\text{model}}$ = 200 | 60.91 | 61.23 | 44.37 | 59.55 |
> | mix, $w_{\mathcal{R}\_\text{human}} / w_{\mathcal{R}_\text{model}}$ = 300 | 63.64 | 63.12 | 43.43 | 63.00 |
> | mix, $w_{\mathcal{R}\_\text{human}} / w_{\mathcal{R}_\text{model}}$ = 400 | 66.72 | 66.32 | 41.83 | 64.47 |
> | mix, $w_{\mathcal{R}\_\text{human}} / w_{\mathcal{R}_\text{model}}$ = 500 | 62.85 | 63.06 | 43.46 | 65.00 |
> | mix, $w_{\mathcal{R}\_\text{human}} /w_{\mathcal{R}_\text{model}}$ = 600 | 56.30 | 56.07 | 46.95 | 60.25 |
>
> It shows that when trained with only $\mathcal{R}\_\text{human}$ or $\mathcal{R}\_\text{model}$, the selection model's performance decreases compared to the stratified mixture of both preference data. This indicates that while high-quality, the limited amount of $\mathcal{R}\_\text{human}$ annotations is insufficient for robust data selection model training. In comparison, stratified mixing of $\mathcal{R}\_\text{human}$ and $\mathcal{R}\_\text{model}$ significantly improves the model's performance compared to using only $\mathcal{R}\_\text{model}$. The best performance is achieved with a $w_{\mathcal{R}\_\text{human}}/w_{\mathcal{R}\_\text{model}}$ ratio of 400, demonstrating the importance of re-balancing human annotation with model-rated data.
>
> These findings strongly support our approach of mixing $\mathcal{R}\_\text{human}$ with $\mathcal{R}\_\text{model}$. This mixture effectively balances the high-quality but limited clinician annotations with scalable model-based annotations, resulting in a more robust and accurate selection model while minimizing clinician annotation effort.

---

> ### Author Rebuttal · Authors · 2024-08-17
>
> > **Q8: Explain the dips and peaks in Figure 2: are they just random performances?**
>
> **A**: The performance curve in Figure 2 exhibits an overall decreasing trend as more data from the ranking list‘s tail is included. The observed dips and peaks aren't random but rather a result of non-uniform data distribution and varying data patterns, possibly be attributed to:
> - Data quality clusters: the dataset likely contains clusters of varying quality, leading to non-linear changes in performance as these clusters are included or excluded at different thresholds.
> - Data distribution patterns: certain patterns or characteristics in the data may influence the selection model's performance, causing localized fluctuations in the curve.
>
> While effective, our selection model may respond differently to various data distributions, contributing to the observed peaks and dips. Despite these fluctuations, the overall decreasing trend remains the key takeaway, indicating that our approach effectively identifies and prioritizes higher-quality data.
>
> Regarding the thresholds chosen for our experiments, we selected the top 50% (instead of 30%) and the top 80% because they demonstrate similar F1 scores and precision when compared to human annotations. This allows us to understand the influence of the scaling law in model training. Additionally, we included the top 10% threshold as it represents a subset of data that strikes a balance between quality and quantity to help better understand the effectiveness of data selection.
>
> > **Q9: Limitations should be part of the paper and not be in the appendix.**
>
> **A**: Thank you for the suggestion! We'll restructure to include the main issues in the final version.
>
> > **Q10: the potential bias of GPT-4V-based evaluation for Sec 4.3 and are there human ratings**
>
> **A**: Thank you for your inquiry! Please refer to our global response under **Q2**.
>
> > **Q11: How are the numbers in Table 2 calculated.**
>
> **A**: The scores in Table 2 are relative scores calculated following LLaVA-Med. When comparing a response generated by a model to a reference answer from GPT-4V, the evaluator is asked to provide scores for both, and a relative score is then calculated by dividing the model's score by the reference's score $S_\text{model}$ / $S_\text{reference}$. The numbers are these relative scores, e.g., a score of 65 indicates that the compared model distilled 65% of GPT-4V's capability. Regarding "Question Types" and "Domains" on the columns, they represent two different ways to categorize the questions in the corresponding dataset. "Overall" is a weighted average across all categories.
>
> > **Q12: Typo in a reference**
>
> **A**: Thank you for catching this oversight! It should be NeurIPS 2023. :< We apologize for this mistake and will update to reflect the correct information.
>
> > **Q13: More information about the VQA benchmarks is desirable**
>
> **A**: Thanks for the suggestion. In the current version of our paper, we prioritized detailing the key data-centric techniques of our framework to facilitate easy replication of our data curation process. Given the established nature of the VQA benchmarks we used, we initially included fewer details about these evaluation datasets. We recognize the value of additional context for readers and will make sure to incorporate it in our final version.
>
> > **Q14: Similar to Sec 4.3, GPT-4V evaluation issue with the win-rate performance**
>
> **A**: Note that we have included a comprehensive set of metric calculations (objective metrics such as accuracy and recall) on three VQA benchmarks in Table 3. These results are presented alongside literature-reported outcomes from representative state-of-the-art methods, providing a broad context for our model's performance. Regarding the GPT-4V-based win-rate evaluation, we specifically employed this method for model ablations. The primary goal was to gain insights into the effectiveness of different modules within our proposed framework. We acknowledge that GPT-4V-based evaluation could introduce some bias. However, we believe that the horizontal comparison can still reveal valuable insights into the relative performance differences between model variants.
>
> > **Q15: Why are not the same questions used for both models in the case studies?**
>
> **A**: Thank you for the question. It is important to clarify that we have different approaches for the several case study figures in our paper. In Figure 4, the case study compares the generated instruction-response pairs from our method and LLaVA-Med. Both the questions and responses shown in the left and right panels are model-generated instruction-response data from their respective methods, intended for finetuning a language-vision model. Therefore, they are intentionally different: we not only need to compare the quality of the generated answers, but also the value and comprehensiveness of the generated questions. In contrast, the case studies on downstream tasks presented in Appendix Figure 9 (open-ended visual chat) and Figure 10 (VQA) use questions sourced from benchmark datasets. Consequently, in these figures, the questions are indeed the same for both models being compared, providing a direct comparison of how different models respond to identical prompts from established benchmarks.

---

> > ### Author Rebuttal · Authors · 2024-08-17
> >
> > > **Q16: Case studies are not part of the main paper and were the cases selected; How reliable is the evaluation and any human evaluations**
> >
> > **A**: We included one randomly selected case study of our generated instruction data in Figure 4. The primary purpose is to illustrate the system's input and output for better understanding rather than serve as a primary evaluation. For a quantitative evaluation, we rely on the downstream performance of instruction-tuned models that have been fine-tuned with our data, which provides a more robust and objective assessment of the dataset's quality and effectiveness. The case studies presented in Figures 9 and 10 of the appendix serve a complementary role to the quantitative metrics shown in Table 2, Table 3, and Figure 3. The case studies are reviewed and highlighted by clinician readers and aim to offer readers a concrete illustration of the model's responses compared to an existing model.
> >
> > > **Documentation: instructions on how to get the images to the annotations and the full dataset viewer on Huggingface**
> >
> > **A**: We appreciate your feedback and have made several improvements to the Huggingface dataset documentation.  We have resolved the issues with the full dataset viewer mode, updated README files to include image URLs, and enhanced the markdown documentation to include detailed dataset information. All the complete datasets are now available to download so readers can explore locally as well.
> >
> > Please let us know if we have addressed your concerns. Thank you!

---

> ### Comment · Reviewer_s4zq · 2024-08-19
> **thx for the rebuttal**
>
> I thank the authors for their very detailed replied and improvements. I hope these detailed information can be added to the final version as promised and thus I increased my score by 2 to 7.
>
> I'm still not fully convinced by some details like using majority vote for disagreement, ranking loss seems to similar to proposed loss and the GPT evaluation still might have issues.  However, I recognize the arguments of the authors and agree that it can be done this way.

---

> > ### Author Response · Authors · 2024-08-19
> >
> > We are pleased that we have addressed some of your concerns in our rebuttal. Thank you for raising the score, and we will ensure that the discussions in the rebuttal are properly incorporated into our revised paper!

---

### Official Review · Reviewer_K4A9 · 2024-07-24
**Effective Generating and Selecting Instruction Data for Tuning MLLM in Biomedicine Domain**

**Rating:** 7
**Confidence:** 4
**Clarity:** The paper is well written

**Review:**

The BioMed-VITAL framework provides a significant contribution to the field of biomedical AI by offering a novel approach to aligning visual instruction tuning with clinical expertise.

Quality: The research demonstrates high quality through its rigorous methodology and comprehensive evaluation. The use of clinician preferences in both data generation and selection stages shows a deep understanding of the domain's needs. The performance improvements across multiple benchmarks further attest to the framework's effectiveness.

Clarity: The paper presents its ideas and methodology clearly, making it accessible to researchers in both AI and healthcare fields. The step-by-step approach to data generation and selection is well-explained, facilitating potential replication and extension of the work.
Originality: BioMed-VITAL introduces an innovative paradigm by incorporating clinician preferences into the data pipeline for multimodal foundation models. This approach distinguishes itself from previous methods by explicitly aligning AI with domain expertise, marking a novel direction in specialized AI development.

Significance: The work's significance is evident in its potential impact on biomedical AI applications. By improving performance in open visual chat and medical VQA tasks, it paves the way for more reliable and clinically relevant AI assistants in healthcare. The open-sourcing of datasets and models further amplifies its significance, enabling broader research and development in the field.
The careful annotation process, significant performance improvements, and open resource contribution make this work stand out. However, challenges related to scalability, and Lack of validation on enough large medical multimodal models should be considered for future development and application.

**Strengths:**

Clinician Preference Alignment: The proposed framework uniquely incorporates clinician preferences directly into the data generation and selection processes, ensuring that the resulting datasets are highly relevant to medical professionals' needs.

Significant Performance Improvement: The BioMed-VITAL approach demonstrates a substantial improvement in performance metrics, with an 18.5% relative increase in open visual chat and up to an 81.73% win rate in medical Visual Question Answering (VQA), showcasing the effectiveness of the clinician-aligned tuning.

Comprehensive Methodology: The paper presents a detailed three-stage process, including data generation with demonstrations, data selection with a preference-distilled model, and visual instruction-tuning, which is thorough and replicable.

Open Resource Contribution: By releasing the clinician preference-aligned datasets and instruction-tuned models, the authors contribute valuable resources to the research community, promoting further exploration and innovation in biomedical AI.

**Additional Feedback:**

The authors are encouraged to address the points mentioned in the "Opportunities for Improvement" section. Specifically:

Expand the evaluation to include a wider range of multimodal language models.
Implement measures to mitigate potential biases in the clinician-annotated data, such as ensuring diverse clinician representation in the annotation process.

Addressing these issues would significantly strengthen the study and could lead to a reconsideration of the overall score.

**Correctness:**

The methodology employed in developing the BioMed-VITAL-instructions dataset demonstrates scientific rigor and validity. The evaluation protocols and experimental design exhibit overall appropriateness and align with standard practices in the field. However, the study's robustness could be further enhanced by expanding the range of Multimodal Large Language Models (MLLMs) included in the evaluation process. This expansion would provide a more comprehensive assessment of the framework's effectiveness across diverse model architectures and capabilities.

**Documentation:**

The BioMed-VITAL framework introduces a novel dataset of instruction-following examples, curated through a process that aligns with clinical expertise. While this work does not present a traditional comparative analysis as seen in other QA dataset papers, it does offer a unique perspective on the integration of clinician preferences within the domain of biomedical multimodal AI.

**Ethics:**

I do not see any ethical concerns with the submission.

**Limitations:**

The authors adequately address the limitations of their study and potential negative societal impacts of their work. Their discussion of these aspects is clear and comprehensive, demonstrating a balanced view of the research's scope and implications.

**Opportunities For Improvement:**

While the paper presents a robust framework, there is a need for more models to be trained and evaluated to fully understand the clinician preference-aligned instruction data's capabilities and limitations. This would help validate the framework's generalizability across different model architectures and sizes.
The BioMed-VITAL framework may inherit biases present in the clinician-annotated data, which could affect the model's fairness and accuracy in diverse medical contexts. These biases could stem from the clinicians' backgrounds, experiences, and the patient populations they typically serve. To mitigate this, ensuring a diverse range of clinicians contribute to the data annotation process, representing various specialties, geographic locations, and patient populations, would be crucial for enhancing the model's robustness and fairness across different demographic groups and medical conditions.

**Relation To Prior Work:**

Yes, the work clearly discusses how it differs from previous contributions by introducing a new data-centric framework that aligns clinician preferences with the generation and selection of instruction-following datasets for biomedical visual instruction tuning. This approach is novel in its incorporation of expert knowledge directly into the dataset curation process, leading to improved model performance in specialized biomedical tasks.

**Summary And Contributions:**

This paper introduce BioMed-VITAL which aligns instruction datasets with domain expertise for biomedical domain, and the contribution as follows:
1. Introducing a data-centric framework for generating and selecting clinician preference-aligned instruction data
2. Proposing a paradigm involving clinician preference in generation and a preference-based data selection model
3. Improving LLaVA model performance in open visual chat (18.5% relative) and biomedical VQA (up to 81.73% win rate)
4. Open-sourcing 80K clinician preference-aligned instruction datasets, along with instruction-tuned models

---

> ### Author Rebuttal · Authors · 2024-08-17
>
> Thank you for your review! We appreciate your positive and thoughtful feedback. Please review our additional results and discussions on the points raised below.
>
> > **Q1: Results of a wider range of multimodal language models evaluated to validate the framework's generalizability across different model architectures and sizes**
>
> **A**: Thank you for your valuable suggestions! We have conducted additional experiments with recent language-vision models, including LLaVA-OneVision [1] and InternVL-1.5 [2], trained on our generated instruction data and evaluated on three VQA benchmarks. The results are summarized in the table below:
>
> | **Language-Vision Model** | **SFT** | **VQA-RAD (Open)** | **VQA-RAD (Closed)** | **SLAKE (Open)** | **SLAKE (Closed)** | **PathVQA (Open)** | **PathVQA (Closed)** |
> |------------------------|-----|----------------|------------------|--------------|----------------|----------------|------------------|
> | LLaVA | No | 50.00 | 65.07 | 78.18 | 63.22 | 7.74 | 63.20 |
> | LLaVA | Yes (w/ LLaVA-Med Dataset) | 61.52 | 84.19 | 83.08 | 85.34 | 37.95 | 91.21 |
> | LLaVA | Yes (w/ Ours) | 63.46 | 84.71 | 85.41 | 87.26 | 38.96 | 92.39 |
> | LLaVA-OneVision | No | 51.26 | 69.49 | 76.45 | 63.46 | 8.24 | 67.50 |
> | LLaVA-OneVision | Yes (w/ LLaVA-Med Dataset) | 62.37 | 78.68 | 83.54 | 86.29 | 37.44 | 92.06 |
> | LLaVA-OneVision | Yes (w/ Ours) | 62.43 | 83.82 | 85.59 | 87.02 | 39.25 | 92.18 |
> | InternVL-1.5 | No | 49.31 | 60.29 | 76.63 | 62.74 | 9.37 | 63.25 |
> | InternVL-1.5 | Yes (w/ LLaVA-Med Dataset) | 61.22 | 81.25 | 84.24 | 81.97 | 37.14 | 91.53 |
> | InternVL-1.5 | Yes (w/ Ours) | 62.38 | 84.92 | 86.75 | 83.17 | 39.20 | 92.65 |
>
> Key findings:
> - Fine-tuning with our instruction data significantly improves model performance across all three benchmarks, demonstrating the effectiveness of our framework in generating helpful training data.
> - Our approach shows consistent improvement in both open and closed categories compared to fine-tuning with LLaVA-Med datasets, highlighting the benefits of clinician alignment.
> - Performance gains are observed across different model architectures (LLaVA and LLaVA-OneVision), indicating the generalizability of our approach.
>
> In addition, we conducted experiments using models of various sizes (7B and 13B parameters). The results are presented in the table below:
> | Model     | Size | VQA-RAD (Open) | VQA-RAD (Closed) | SLAKE (Open) | SLAKE (Closed) | PathVQA (Open) | PathVQA (Closed) |
> |-----------|------|----------------|------------------|--------------|----------------|----------------|------------------|
> | LLaVA     | 7b   | 50.00          | 65.07            | 78.18        | 63.22          | 7.74           | 63.20            |
> | LLaVA     | 13b  | 52.23          | 63.23            | 76.59        | 64.42          | 8.82           | 66.32            |
> | LLaVA-Med | 7b   | 61.52          | 84.19            | 83.08        | 85.34          | 37.95          | 91.21            |
> | LLaVA-Med | 13b  | 64.58          | 77.94            | 84.97        | 85.58          | 38.82          | 92.39            |
> | Ours      | 7b   | 63.46          | 84.71            | 85.41        | 87.26          | 38.96          | 92.39            |
> | Ours      | 13b  | 64.88          | 84.55            | 87.82        | 86.54          | 39.71          | 91.41            |
>
> It shows that our framework consistently outperforms both LLaVA and LLaVA-Med across most tasks and model sizes, particularly notable in the open-ended questions for all datasets. This finding underscores the generalizability of our proposed framework across models with various sizes.
>
> *[1] Li, B., Zhang, Y., Guo, D., Zhang, R., Li, F., Zhang, H., ... & Li, C. (2024). LLaVA-OneVision: Easy Visual Task Transfer. arXiv preprint arXiv:2408.03326.*
>
> *[2] Chen, Z., Wu, J., Wang, W., Su, W., Chen, G., Xing, S., ... & Dai, J. (2024). Internvl: Scaling up vision foundation models and aligning for generic visual-linguistic tasks. In Proceedings of the IEEE/CVF Conference on Computer Vision and Pattern Recognition (pp. 24185-24198).*

---

> ### Author Rebuttal · Authors · 2024-08-17
>
> > **Q2: Inherit biases of the clinician-annotated data and measures to mitigate them**
>
>
> **A**:  Thank you for your insightful suggestion! We acknowledge the potential biases in our clinician-annotated data and agree that involving a more diverse range of clinicians would be a valuable expansion.
> To explore this direction, we conducted two proxy studies:
>
> - **Study on Preference Annotation with Diverse Backgrounds**: We augmented various specialty roles (e.g., "You are a specialist in Chest X-Ray / MRI / Histology / Gross Pathology / CT Scan") to GPT4V prompts as a proxy for clinician preference annotation with different specialties. We then calculated the inter-annotator agreement in the clinician preference annotation process among these simulated specialists using Fleiss' kappa on the randomly selected 100 examples. The resulting kappa value was 0.728, which is comparable to the 0.736 kappa value obtained from the annotations of our three annotators from Massachusetts General Hospital's Radiology department. This minimal difference suggests that the preference annotation task is relatively straightforward and not significantly influenced by the annotator's specific clinical background. We conjecture that consensus is easily reached because question-answer pairs labeled as lower quality generally include flaws or errors.
> - **Study on Instruction Data Generation with Diverse Backgrounds**: We incorporated five different specialties, each corresponding to the five image modalities in our datasets, into our instruction data generation prompts. We also included a few-shot examples of the corresponding category and generated instruction data for 500 input images. To measure the diversity of the generated instruction data, we calculated the average cosine distance of their semantic embeddings. We found that the distance of the generated instruction data increased from 0.3029 (without specialties) to 0.3346 (with specialties), indicating a 10.5% diversity improvement. This suggests that using data generator prompts with roles from diverse backgrounds leads to improved instruction data diversity. It points towards a promising direction for future research to expand our framework.
>
> It's worth noting that our current framework already incorporates some designs to increase the diversity of instruction data. We designed the diverse few-shot demonstration selection and randomly chose 2 samples for each of the 5 image modalities during each data generation. They are intended to increase the diversity of the generated instruction data and ultimately enhance the performance of models trained on it.
>
> We appreciate your constructive feedback on involving clinicians with various backgrounds. We will incorporate these insightful discussions and preliminary insights in the paper to emphasize future directions for data diversity and bias mitigation.
>
> Please let us know if we have addressed your concerns. Thank you!

---

### Official Review · Reviewer_c7Cq · 2024-07-25
**Paper Review**

**Rating:** 7
**Confidence:** 5
**Clarity:** Yes, it's well written.

**Review:**

The work has significant implications for the field of biomedical AI. By demonstrating the effectiveness of clinician-aligned data generation and selection, the paper sets a new standard for creating high-quality, domain-specific datasets. The availability of these resources for public use further amplifies the impact of this research.

**Strengths:**

Contributions:

1. BioMed-VITAL Framework: Introduction of a data-centric framework that incorporates clinician preferences into the generation and selection of instruction-following data for biomedical visual instruction tuning.

2. Improved Model Performance: Demonstrated significant improvements in model performance on open visual chat and medical VQA tasks, with a relative improvement of 18.5% in open visual chat and a win rate of up to 81.73% in medical VQA benchmarks.

3. Release of Resources: Provision of 80K clinician preference-aligned instruction-following datasets and the models tuned on these datasets, available for public use at BioMed-VITAL.

**Additional Feedback:**

N/A

**Correctness:**

The claims made in the submission are well-supported by empirical evidence. The dataset is constructed in a sound manner, and the evaluation methods and experiment design are appropriate and correctly executed. The study provides clear and detailed explanations of the methodologies used, ensuring the reproducibility of results.

**Documentation:**

Yes, it's sufficient.

**Limitations:**

The authors have adequately addressed the limitations of their work, particularly the reliance on clinician-annotated data and the challenges associated with it. They propose potential solutions, such as using a mixture of human and model-based preferences to mitigate the scalability issue. The study does not present any immediate negative societal impacts; rather, it aims to improve the quality and applicability of AI models in clinical settings, which could have positive implications for patient care.

**Opportunities For Improvement:**

1. Scalability: Exploring methods to scale the clinician annotation process or leveraging semi-supervised approaches to reduce the dependency on expert input.

2. Generalization: Investigating the application of the BioMed-VITAL framework to other specialized domains to evaluate its generalizability and adaptability.

3. Continous Integration with New Models: One important aspect of benchmarks is the continuous integration of new models and the ability to facilitate comparisons. It is recommended to integrate this into benchmarks toolkits like lmms-eval for ease of use with the community. Additionally, creating a leaderboard to compare the latest results is also recommended.

**Relation To Prior Work:**

Yes, it's clearly discussed.

**Summary And Contributions:**

The paper introduces a novel framework, BioMed-VITAL, designed to enhance the performance of multimodal foundation models in the biomedical domain. The framework integrates clinician preferences into the data generation and selection processes to create high-quality, domain-specific instruction-following datasets. This approach significantly improves the model’s performance in open visual chat and medical visual question answering (VQA) tasks. The study demonstrates that incorporating expert preferences can produce more clinically relevant outputs, thereby enhancing the practical utility of these models in real-world biomedical applications.

---

> ### Author Rebuttal · Authors · 2024-08-17
>
> Thank you for your review. We appreciate your positive feedback! Please see our responses regarding the several opportunities for improvement below.
>
> > **Q1: Exploring methods to scale the clinician annotation process or leveraging semi-supervised approaches to reduce the dependency on expert input.**
>
> **A**: Yes, we agree that exploring methods to scale the clinician annotation process is a promising direction for future work. We appreciate your acknowledgment of our proposed mixture approach, combining human and model-based preferences to address the scalability issues. One possible way is to create proxy agents with various specialties as prompts to increase the diversity of automated clinical data annotation. This approach could potentially enhance the model's performance across different medical subfields. Another approach could be developing automated methods to pre-screen or pre-annotate data, reducing the workload on human clinicians and allowing them to focus on more complex or ambiguous cases. Also, investigating novel semi-supervised training objectives in specialized domains like biomedicine could help reduce the dependency on expert input. We appreciate your raising this insightful discussion, and we will make sure to include more details in the paper.
>
> > **Q2: Investigating the application of the BioMed-VITAL framework to other specialized domains to evaluate its generalizability and adaptability.**
>
> **A**: Thank you for this insightful suggestion! We agree that investigating the generalizability of the BioMed-VITAL framework to other specialized domains is a valuable direction for future research. While our current focus is on biomedical vision-language models, the core techniques in our data-centric framework are designed to be adaptable, allowing researchers and practitioners in different fields to create high-quality instruction training datasets tailored to their specific needs, especially when they want to effectively distill expert preferences with only a few human expert annotations. We are particularly interested in seeing how the framework performs in other specialized domains that share similar challenges. Such investigations would not only validate the framework's versatility but also potentially lead to improvements and refinements in the methodology itself.
>
> > **Q3: Continuous integration of new models and facilitate comparisons**
>
> **A**: Thank you for your valuable suggestions! We have improved the usability of our curated training datasets by enriching our markdown readme documents and providing a data viewer on our Huggingface dataset page. Readers can now fully explore by downloading the complete datasets. Please check our updated Huggingface page for new information! We also agree with the suggestion to facilitate continuous integration and community accessibility. We will establish a living leaderboard to showcase and compare the latest results, as well as an interactive demo to allow users to interact with the model and see its capabilities firsthand.
>
> Thanks again for your valuable feedback! Please let us know if you have any further questions.

---

### Official Review · Reviewer_wFxu · 2024-07-25
**Dataset released as a by-product of the model**

**Rating:** 6
**Confidence:** 4
**Clarity:** This paper is well written.

**Review:**

Please see Strengths and Opportunities For Improvement.

**Strengths:**

+ This work introduces a data-centric framework called BioMed-VITAL, which generates and selects instruction-following data aligned with clinician preference for visual instruction tuning.
+ This paper releases 80K clinician preference-aligned instruction-following datasets as by-products of the core contribution, i.e., the models instruction-tuned based on the datasets.

**Additional Feedback:**

N/A

**Correctness:**

The dataset is only released as a by-product of the model. In addition, there lack of details to know whether the dataset is constructed in a sound way. Please refer to Q4-7 in Opportunities For Improvement.

**Documentation:**

The details of data collection can be improved. Please see Q4-Q7 in Opportunities For Improvement. The documentation, hosting, licensing, and maintenance plan are provided in Suppl.

**Ethics:**

No ethical concerns

**Limitations:**

The authors have discussed the limitations in Supplementary Material.

**Opportunities For Improvement:**

1. This paper focuses more on the model design than on the dataset/benchmark. Although this paper releases an 80K clinician preference-aligned instruction-following dataset, most details on the dataset construction and collection are in the Supplementary Material, with only three paragraphs in Sec. 3.1 describing the dataset. Thus, I am not sure whether this paper should be evaluated according to the criteria of Dataset and Benchmark Track or not.

2. If we follow the criteria of Dataset and Benchmark Track, the main body of this paper provides too few details on the dataset construction and collection. Additionally, the main body does not present the results of a variety of SoTA methods trained and evaluated on the proposed dataset. Without these results, we cannot know whether other SoTA methods can benefit from the proposed dataset or not.

3. If we consider this paper as a methodology one, the novelty of the proposed model is minor. To me, the major contribution is collecting a dataset to improve the model performance, while the training scheme of the selection model (e.g., pairwise classification or data selection) and instruction-tuning in Sec. 3 are with minor novelty.

4. How many clinicians participated in the dataset generation, how to ensure the quality of the model-generated and clinician-chosen data?

5. The M (Ln. 110), and N (Ln. 95) are hyper-parameters or not? How to set their values?

6. In Ln. 214, simply using GPT-4V as the evaluator can be biased. Sampling some responses for clinician checking can make the GPT-4V-based results more reliable. That is, please consider evaluating the reliability of GPT-4V-based results.

7. Ln. 112, why are two candidate responses provided rather than more candidates?

**Relation To Prior Work:**

The contributions are presented in Sec. 1 and the differences between this work and previous ones are discussed in Sec. 2.

**Summary And Contributions:**

This paper introduces a data-centric framework named Biomedical Visual Instruction Tuning with Clinician Preference Alignment (BioMed-VITAL) for biomedical visual instruction tuning. The BioMed-VITAL framework generates and selects instruction-following data aligned with clinician preference. Experimental results show that the BioMed-VITAL model tuned with the generated instruction-following data can achieve a significant improvement in open visual chat and medical VQA.

---

> ### Author Rebuttal · Authors · 2024-08-17
>
> We would like to thank the reviewer for the useful feedback. Please see our response to the raised questions in the response below.
>
> > **Q1: Most dataset details are in the supplementary. Should it be evaluated as Dataset and Benchmark Track**
>
> **A**: Our work is to establish a data-centric framework for curating training data for the vision-language model for biomedical models. It fits into the following categories in the call for paper:
> - Data-centric AI methods and tools, e.g., to measure and improve data quality or utility, or studies in data-centric AI that bring important new insight.
> - Advanced practices in data collection and curation are of general interest even if the data itself cannot be shared.
>
> The dataset details are essentially Section 3 in our paper, including the instruction data construction (Section 3.1) and sample selection (Section 3.2). We use raw PMC-15 datasets as input (described in Section 4.1) and produce instruction-response pairs for language-vision model training for biomedical domains. Case studies are in Figure 4 and Appendix Figure 8 to illustrate the output instruction data. We have uploaded the complete datasets from our framework to Huggingface with documentation and a full dataset preview. Users can also explore the datasets by downloading the complete datasets.
>
> Due to space limitations, we focused on the technical details of our data-centric framework in the submission so that others could easily replicate the data curation or create instruction data for new scenarios. We appreciate your comments and will ensure to highlight more dataset information in the final version.
>
> > **Q2: The main body of this paper provides too few details on the dataset construction and collection; results of other SoTA methods trained and evaluated on the proposed dataset**
>
> **A**: For dataset construction details, please see our clarification under **Q1**. We appreciate your suggestion on other models and have conducted experiments with recent SoTA language-vision models, LLaVA-OneVision [1] and InternVL-1.5 [2], trained on our generated instruction data and evaluated on three VQA benchmarks. The results are summarized in the table below:
>
> | Language-Vision Model | SFT | VQA-RAD (Open) | VQA-RAD (Closed) | SLAKE (Open) | SLAKE (Closed) | PathVQA (Open) | PathVQA (Closed) |
> |------------------------|-----|----------------|------------------|--------------|----------------|----------------|------------------|
> | LLaVA | No | 50.00 | 65.07 | 78.18 | 63.22 | 7.74 | 63.20 |
> | LLaVA | Yes (w/ LLaVA-Med Dataset) | 61.52 | 84.19 | 83.08 | 85.34 | 37.95 | 91.21 |
> | LLaVA | Yes (w/ Ours) | 63.46 | 84.71 | 85.41 | 87.26 | 38.96 | 92.39 |
> | LLaVA-OneVision | No | 51.26 | 69.49 | 76.45 | 63.46 | 8.24 | 67.50 |
> | LLaVA-OneVision | Yes (w/ LLaVA-Med Dataset) | 62.37 | 78.68 | 83.54 | 86.29 | 37.44 | 92.06 |
> | LLaVA-OneVision | Yes (w/ Ours) | 62.43 | 83.82 | 85.59 | 87.02 | 39.25 | 92.18 |
> | InternVL-1.5 | No | 49.31 | 60.29 | 76.63 | 62.74 | 9.37 | 63.25 |
> | InternVL-1.5 | Yes (w/ LLaVA-Med Dataset) | 61.22 | 81.25 | 84.24 | 81.97 | 37.14 | 91.53 |
> | InternVL-1.5 | Yes (w/ Ours) | 62.38 | 84.92 | 86.75 | 83.17 | 39.20 | 92.65 |
>
> Key findings:
> - Fine-tuning with our instruction-following dataset can significantly improve model performance across all three benchmarks, demonstrating the effectiveness of our framework in generating helpful training data.
> - Our approach shows consistent improvement in both open- and closed-category questions compared to fine-tuning with LLaVA-Med datasets, highlighting benefits with clinician alignment.
> - Performance gains are observed across different model architectures (LLaVA and LLaVA-OneVision), which indicates the generalizability of our framework.
>
> It's important to note that while our primary focus is on biomedical vision-language models, the core techniques within our data-centric framework are designed to be adaptable. This allows researchers in various fields to create high-quality instruction data that meets their specific requirements.
>
> *[1] Li, B., Zhang, Y., Guo, D., Zhang, R., Li, F., Zhang, H., ... & Li, C. (2024). LLaVA-OneVision: Easy Visual Task Transfer. arXiv preprint arXiv:2408.03326.*
>
> *[2] Chen, Z., Wu, J., Wang, W., Su, W., Chen, G., Xing, S., ... & Dai, J. (2024). Internvl: Scaling up vision foundation models and aligning for generic visual-linguistic tasks. In Proceedings of the IEEE/CVF Conference on Computer Vision and Pattern Recognition (pp. 24185-24198).*
>
> > **Q3: The novelty of the proposed model is minor if considered as a methodology paper**
>
> **A**: Our work addresses a critical, domain-specialized, and under-explored challenge: introducing a data-centric framework that incorporates domain experts’ preferences into the generation and selection of instruction-following data for specialized visual instruction tuning. Our contributions extend beyond the dataset itself to include novel data-centric technical approaches that integrate clinician preferences into the process:
> - a strategic sampling for diverse and dynamic few-shot demonstration selection in data generation
> - a novel data mixture strategy to mitigate the scalability issue for training a data selection model aligned with clinician preferences, using minimal expert annotations to effectively capture their preference.
> - an adaptive mechanism for controlling the contribution of clinicians and model-generated preference data and effective techniques to determine thresholds for the best data selection
>
> In our paper, we empirically demonstrated that these techniques are effective and can provide insights into data curation for specialized AI applications. Besides, our framework is designed to be adaptable, facilitating replication and extension to other scenarios beyond improving model performance in a specific setting.

---

> > ### Comment · Reviewer_wFxu · 2024-08-21
> >
> > Many thanks for the feedback. It is good to see the results of other SoTA methods which verifies the contribution of the proposed datasets, and implicitly shows the effectiveness of the overall proposed framework. Although I still have concerns on the novelty of the proposed method (either the strategic sampling or data mixture), a data-centric framework for curating training data is indeed crucial for clinical applications and the datasets can contribute to the community. Therefore, I will upgrade the rating to 6.
> >
> > - Please consider evaluating the sensitivity of the number of clusters M.
> > - Please ensure to highlight more dataset information and include the answers to the other questions in the final version.

---

> > ### Author Response · Authors · 2024-08-23
> >
> > Thank you for your thoughtful feedback and for upgrading the rating. We greatly appreciate your recognition of the importance of our data-centric framework for clinical applications and the contribution of our datasets to the community. We have carefully noted all suggestions and will ensure to include discussions about them in the camera-ready version and further investigate them in our future work. Once again, thank you for your constructive comments!

---

> ### Author Rebuttal · Authors · 2024-08-17
>
> > **Q4: How many clinicians participated in the annotation and how to ensure the quality of the clinician-chosen and model-generated data**
>
> **A**: Thank you for the question. Three MDs and PhDs from Massachusetts General Hospital's Radiology participated in our annotation. Each clinician was given 300 carefully selected questions with two possible answers and asked to choose the preferred one. Out of the 300 samples, agreement happens in most cases (241 out of 300), meaning the clinicians made the same choice. Disagreement occurred in a small number of cases (59 out of 300) where usually the two answers were very similar in quality and difficult to distinguish. In these cases, we used majority voting to make the final decision. The Fleiss' kappa on the three clinician annotations is 0.736, indicating a good agreement among the three annotators.
>
> There are two kinds of model-generated data in our framework: model-generated instruction data and model-generated preference ratings. For model-generated instruction data, we developed a data selection model in Stage 2 (Section 3.2) to ensure that only high-quality data aligned with clinicians' preferences are included in the final dataset for instruction tuning. For the model-generated preference ratings, models are provided with criteria written by clinicians to distill their preferences. The clinicians then review the model-ranked scores and adjust the prompts accordingly. Additionally, we incorporate dynamic sample weights in the preference mixing during training to strike a better balance between data scalability and quality. Section 4.2 Table 1 illustrates the results to determine the optimal sample weights.
>
> > **Q5: Are the M (Ln. 110), and N (Ln. 95) hyper-parameters; how their values were set**
>
> **A**: M and N are values that can be adjusted based on the desired dataset size and diversity. N is the total number of image-text samples used as input for our data curation pipeline. It's a scalable value that represents sample sizes: e.g., Our generated 80K dataset uploaded to the Huggingface [dataset](https://huggingface.co/datasets/mao1207/BioMed-VITAL-instructions) page. Users can replicate by following the data curation details and create more as they want. M is the number of clusters used in K-Means clustering for sample diversity. For simplicity, we set M to 60 in our experiment.
>
> > **Q6: The reliability of GPT-4V as the evaluator; sampling some response for clinician checking**
>
> **A**: We fully recognize your concern about the GPT-based performance evaluation. Actually, tasks like open-ended visual chat can be challenging to evaluate, while traditional NLP metrics are not sufficient to capture the semantic and higher-order abstract of the text. Recent studies have explored the use of large language models such as GPT as evaluators, demonstrating their greater resilience compared to previous metric-based approaches [1,2,3,4,5].
>
> To validate the reliability of GPT-4V-based evaluation in our experiment, we conducted a quantitative correlation analysis: we randomly selected a subset of 100 samples for clinician rating and determined the Pearson correlation coefficient between human and GPT-4V-generated relative scores. The strong correlation of 0.8322 demonstrates consistent ratings between humans and GPT-4V, validating the validity of using GPT-4V as a proxy evaluator in the task of open-ended visual chat to reduce human effort.
>
> It's important to note that our evaluation strategy extends beyond just GPT-4V-based assessments. Clinicians have been actively involved in verifying the quality of the generated data at each stage during the development of our framework. This comprehensive approach ensures a thorough evaluation process. In Task 1 (open-ended visual chat), we employed GPT-4V as a proxy evaluator, corroborated by the correlation study with clinician ratings.
> The correlation results of 0.8322 demonstrate that GPT-based evaluation is in correspondence with human evaluation. Task 2 involved three VQA benchmarks, and we utilized a multi-faceted evaluation strategy. This included objective assessment with quantitative metrics ( accuracy for closed-set questions and recall for open-set questions), model-based ratings (win rate), and qualitative validation case studies reviewed by clinicians. By combining these evaluations, we've provided a comprehensive evaluation of our proposed framework's performance across a wide range of scenarios.
>
> *[1] Chan, D., Petryk, S., Gonzalez, J., Darrell, T., & Canny, J. (2023). CLAIR: Evaluating Image Captions with Large Language Models. EMNLP, 13638–13646.*
>
> *[2] Chiang, C.-H., & Lee, H.-y. (2023). Can Large Language Models Be an Alternative to Human Evaluations? ACL, 15607–15631.*
>
> *[3] Ging, S., Bravo, M. A., & Brox, T. (2024). Open-ended VQA benchmarking of Vision-Language models by exploiting Classification datasets and their semantic hierarchy. ICLR.*
>
> *[4] Liu, Y., Iter, D., Xu, Y., Wang, S., Xu, R., & Zhu, C. (2023). G-Eval: NLG Evaluation using Gpt-4 with Better Human Alignment. EMNLP, 2511–2522.*
>
> *[5] Mañas, O., Krojer, B., & Agrawal, A. (2024). Improving Automatic VQA Evaluation Using Large Language Models. AAAI, 38(5), 4171-4179.*
>
> > **Q7: Why are two candidate responses provided rather than more candidates (Ln. 112)**
>
> **A**: We agree that more than two candidates could offer a greater variety of responses and potentially better capture human preference. Considering a fixed budget for clinician annotation, we chose to prioritize sample diversity over having more candidates in our annotation process. Therefore, we provided annotators with only two candidates per sample to consider in order to streamline and simplify the annotation process. This method enables us to encompass a broader spectrum of cases with diverse and representative samples.
>
> Please let us know if we have addressed your concerns. Thanks again for your valuable feedback!

---

### Author Rebuttal · Authors · 2024-08-17

We sincerely thank all reviewers for their insightful feedback. Your expert knowledge has helped us strengthen the manuscript significantly. Here, we provide a response to some general questions:

**Common Q1: Additional experiments of language-vision models trained with the dataset from our proposed framework**

**A**: We appreciate the suggestion to train other SoTA models with the dataset generated from our proposed framework to investigate generalizability. We conducted additional experiments with recent SoTA language-vision models, including LLaVA-OneVision [1] and InternVL-1.5 [2], trained on our generated instruction data, and evaluated on three VQA benchmarks. The results are summarized below:

| **Language-Vision Model** | **SFT** | **VQA-RAD (Open)** | **VQA-RAD (Closed)** | **SLAKE (Open)** | **SLAKE (Closed)** | **PathVQA (Open)** | **PathVQA (Closed)** |
|------------------------|-----|----------------|------------------|--------------|----------------|----------------|------------------|
| LLaVA | No | 50.00 | 65.07 | 78.18 | 63.22 | 7.74 | 63.20 |
| LLaVA | Yes (w/ LLaVA-Med Dataset) | 61.52 | 84.19 | 83.08 | 85.34 | 37.95 | 91.21 |
| LLaVA | Yes (w/ Ours) | 63.46 | 84.71 | 85.41 | 87.26 | 38.96 | 92.39 |
| LLaVA-OneVision | No | 51.26 | 69.49 | 76.45 | 63.46 | 8.24 | 67.50 |
| LLaVA-OneVision | Yes (w/ LLaVA-Med Dataset) | 62.37 | 78.68 | 83.54 | 86.29 | 37.44 | 92.06 |
| LLaVA-OneVision | Yes (w/ Ours) | 62.43 | 83.82 | 85.59 | 87.02 | 39.25 | 92.18 |
| InternVL-1.5 | No | 49.31 | 60.29 | 76.63 | 62.74 | 9.37 | 63.25 |
| InternVL-1.5 | Yes (w/ LLaVA-Med Dataset) | 61.22 | 81.25 | 84.24 | 81.97 | 37.14 | 91.53 |
| InternVL-1.5 | Yes (w/ Ours) | 62.38 | 84.92 | 86.75 | 83.17 | 39.20 | 92.65 |

Key findings:
- Fine-tuning with our instruct data significantly improves model performance across all three benchmarks, demonstrating the effectiveness of our framework in generating helpful training data.
- Our approach shows consistent improvement in both open and closed categories compared to fine-tuning with LLaVA-Med datasets, highlighting the benefits of clinician alignment.
- Performance gains are observed across different model architectures, indicating the generalizability and robustness of our approach.

We also conducted experiments using models of various sizes (7B and 13B parameters):

| Model     | Size | VQA-RAD (Open) | VQA-RAD (Closed) | SLAKE (Open) | SLAKE (Closed) | PathVQA (Open) | PathVQA (Closed) |
|-----------|------|----------------|------------------|--------------|----------------|----------------|------------------|
| LLaVA     | 7b   | 50.00          | 65.07            | 78.18        | 63.22          | 7.74           | 63.20            |
| LLaVA     | 13b  | 52.23          | 63.23            | 76.59        | 64.42          | 8.82           | 66.32            |
| LLaVA-Med | 7b   | 61.52          | 84.19            | 83.08        | 85.34          | 37.95          | 91.21            |
| LLaVA-Med | 13b  | 64.58          | 77.94            | 84.97        | 85.58          | 38.82          | 92.39            |
| Ours      | 7b   | 63.46          | 84.71            | 85.41        | 87.26          | 38.96          | 92.39            |
| Ours      | 13b  | 64.88          | 84.55            | 87.82        | 86.54          | 39.71          | 91.41            |

It shows that our framework consistently outperforms both LLaVA and LLaVA-Med across tasks and model sizes, particularly notable in the open-ended questions for all datasets. This underscores our framework's generalizability across model sizes.

*[1] Li, B. et al. "LLaVA-OneVision: Easy Visual Task Transfer." arXiv:2408.03326, 2024*

*[2] Chen, Z. et al. "Internvl: Scaling up vision foundation models and aligning for generic visual-linguistic tasks." CVPR, 2024*


**Common Q2: Reliability of using GPT-4V as the evaluator for open-ended visual chat**

**A**: We recognize the concern about GPT-based evaluation. Open-ended visual chat can be challenging to evaluate, with traditional NLP metrics insufficient to capture the semantic and higher-order abstract text aspects. Recent studies have explored using LLMs as evaluators, demonstrating their greater resilience compared to metric-based approaches [1,2,3].

To validate the reliability of GPT-4V-based evaluation in our experiment, we conducted a quantitative correlation analysis: we randomly selected 100 samples for clinician rating and determined the Pearson correlation coefficient between human and GPT-4V-generated relative scores. The strong correlation of 0.8322 demonstrates consistent ratings, affirming GPT-4V’s validity as a proxy evaluator to reduce human effort.

Also, it's important to note that our evaluation extends beyond just GPT-4V-based assessments. Clinicians have been actively involved in verifying generated data quality at each stage in our framework. In Task 1 (open-ended visual chat), we employed GPT-4V as a proxy evaluator, corroborated by clinician rating correlation study. Task 2 involves three VQA benchmarks, and we utilized a multi-faceted evaluation strategy, including objective assessment with quantitative metrics (accuracy for closed-set and recall for open-set), model-based ratings (win rate), and qualitative validation case studies reviewed by clinicians. Combining these evaluations, we've provided a thorough overview of our framework's performance across a wide range of scenarios.

*[1] Chan, D., Petryk, S., Gonzalez, J., Darrell, T., & Canny, J. (2023). CLAIR: Evaluating Image Captions with Large Language Models. EMNLP*

*[2] Chiang, C.-H., & Lee, H.-y. (2023). Can Large Language Models Be an Alternative to Human Evaluations? ACL*

*[3] Mañas, O., Krojer, B., & Agrawal, A. (2024). Improving Automatic VQA Evaluation Using Large Language Models. AAAI*

Our response to each reviewer has been posted separately. We look forward to interacting with you further during the discussion period!

Warm regards,

Authors of Submission 1823

---

### Author Response · Authors · 2024-08-31

We are grateful for the constructive feedback from the reviewers and are pleased to report that our rebuttal has addressed initial concerns and garnered stronger support for our work. Key points of recognition include:

- Dataset Contribution: Reviewers acknowledged the value of our datasets to the field.
- Framework Effectiveness: Our proposed framework was recognized as effective, with comparisons to state-of-the-art methods that demonstrate its capabilities across tasks.
- Data-Centric Framework: The importance of our framework for curating instruct training data that aligns with clinician preferences.

While some discussion points remain (e.g., novelty and specific methodological choices), reviewers indicated these do not significantly detract from the paper's overall contribution. Given the positive feedback and recognized value of our work, we believe our work offers a vital contribution to the field. We are committed to addressing the remaining minor concerns and incorporating the questions raised during the review process in the final version.

---

### Decision · Program_Chairs · 2024-09-26

**Decision:**

Accept (Poster)

**Comment:**

The reviewers were pleased with the detailed explanation of the proposed algorithm and its thorough experimental validation. Therefore, I recommend to accept it.